# Satellites reveal hotspots of global river extent change

Qianhan Wu[1,2,12], Linghong Ke [3,4,12], Jida Wang [5], Tamlin M. Pavelsky [6], George H. Allen [7], Yongwei Sheng [8], Xuejun Duan[1], Yunqiang Zhu[9], Jin Wu [2], Lei Wang[1], Kai Liu [1], Tan Chen[1], Wensong Zhang[10], Chenyu Fan[1], Bin Yong[3,4] & Chunqiao Song [1,11] ✉

Rivers are among the most diverse, dynamic, and productive ecosystems on Earth. River flow regimes are constantly changing, but characterizing and understanding such changes have been challenging from a long-term and global perspective. By analyzing water extent variations observed from four-decade Landsat imagery, we here provide a global attribution of the recent changes in river regime to morphological dynamics (e.g., channel shifting and anabranching), expansion induced by new dams, and hydrological signals of widening and narrowing. Morphological dynamics prevailed in ~20% of the global river area. Booming reservoir constructions, mostly skewed in Asia and South America, contributed to ~32% of the river widening. The remaining hydrological signals were characterized by contrasting hotspots, including prominent river widening in alpine and pan-Arctic regions and narrowing in the arid/semi-arid continental interiors, driven by varying trends in climate forcing, cryospheric response to warming, and human water management. Our findings suggest that the recent river extent dynamics diverge based on hydroclimate and socio-economic conditions, and besides reflecting ongoing morphodynamical processes, river extent changes show close connections with external forcings, including climate change and anthropogenic interference.

Rivers are one of the most dynamic water cycle components of the earth surface and hold fundamental economic and ecological significance for the development of human societies[1,2], ecosystem sustainability[3], and regional climate[4]. Yet, their natural balance has been threatened by a wide range of anthropogenic stressors and ongoing climate change[5]. With increasing demands for economic and social development, human disturbances in the form of dam construction, aquaculture, and irrigation have resulted in large-scale and rapid transformations of river channels[5]. For instance, many new hydropower stations have been built, with Gigawatt capacity

[1]Key Laboratory of Watershed Geographic Sciences, Nanjing Institute of Geography and Limnology, Chinese Academy of Sciences, Nanjing 210008, China. [2]School of Biological Sciences and Institute for Climate and Carbon Neurality, The University of Hong Kong, Pokfulam Road, Hong Kong, China. [3]College of Hydrology and Water Resources, Hohai University, Nanjing 210098, China. [4]State Key Laboratory of Hydrology-Water Resources and Hydraulic Engineering, Hohai University, Nanjing 210098, China. [5]Department of Geography and Geospatial Sciences, Kansas State University, Manhattan, KS 66506, USA. [6]Department of Earth, Marine and Environmental Sciences, University of North Carolina, Chapel Hill, NC, USA. [7]Department of Geosciences, Virginia Polytechnic Institute and State University, Blacksburg, VA, USA. [8]Department of Geography, University of California, Los Angeles, CA 90095, USA. [9]State Key Laboratory of Resources and Environmental Information System, Institute of Geographic Sciences and Natural Resources Research, Chinese Academy of Sciences, Beijing 100101, China. [10]School of Geography and Ocean Science, Nanjing University, Nanjing 210023, China. [11]University of Chinese Academy of Sciences, Nanjing (UCASNJ), Nanjing 211135, China. [12]These authors contributed equally: Qianhan Wu, Linghong Ke. ✉e-mail: cqsong@niglas.ac.cn

increasing by 55% between 2000 and 2015[5,6], substantially widening the natural wetted river channels. Other human interference[7–9], such as agriculture, deforestation, and wetland destruction, have affected approximately 75% of the ice-free lands[10,11], resulting in radical changes in river morphology (e.g., river fragmentation), sediment flux and ecological balance and diversity (e.g., aquatic species, spread of non-native species). Meanwhile, climate change[12], as evidenced by global warming[13,14], accelerated glacier melting[15] and permafrost thawing[16], and intensified flooding and drought events[17], has further exacerbated the vulnerability and instability of the world's river channels[18]. In response to climate change and human disturbance, the world's rivers show high spatial and temporal variability in flow and sediment regimes, leading to severe environmental, ecological, economic and societal issues[5,19]. These emerging issues call for collaborative, multi-disciplinary and intergovernmental efforts to seek alternatives of anthropogenic stressors and to initiate better management plans for sustainable development, which are built on solid scientific observations and knowledge of how rivers have responded to various forcings.

Earth observation satellites are increasing the feasibility of monitoring large-scale and long-term river channel changes in remote areas with no or limited in situ measurements and regions where sharing in situ data are politically restricted. Recent studies have employed remotely sensed observations to examine global surface water dynamics, but these advances did not include a thorough and necessary distinction of water body types, such as lakes, reservoirs, and rivers, whose changes have different ecological and social ramifications[20–22]. Long-term river flow regime changes on the global scale have not been adequately investigated, partially because identifying and mapping river channels is technically challenging[20,23–25]. A recent study[26] analyzed how monthly and yearly fractional river

extents correlate with the terrestrial water storage components, but their emphasis was river extent variability rather than long-term trends, and their attribution was limited to hydrometeorological factors. Another recent effort by Feng, Gleason[27] investigated temporal variations of river width during 1984–2020 by mapping rivers (average width > 90 m) from the time series of Landsat images. While Feng et al.[27] could not accurately measure river width variations for rivers narrower than ~90 m from Landsat, it would be possible to examine patterns and trends in total inundation extent for many narrower rivers. Moreover, while they examined width variations associated with reservoirs, they did not explicitly separate changes associated with reservoirs from other alterations to river width.

Here, we provide a new framework for quantifying and interpreting multi-decadal river extent changes on large spatial scales through leveraging two major state-of-the-art surface water databases: the Surface Water and Ocean Topography River Database (SWORD)[28,29] and the occurrence change intensity (OCI) map from the Global Surface Water (GSW) database[22] (see Methods). The OCI map documents the change of water inundation frequency on continental surfaces between two epochs (the late 20th century and the early 21st century) based on long-term Landsat images (1984–2018) at 30-m resolution[22]. We categorize the OCI values into five classes: significantly increased (SI), moderately increased (MI), generally stable (GS), moderately decreased (MD), and significantly decreased (SD), to describe frequency changes of different magnitudes (See Methods for the classification). The pattern of frequency changes on the OCI map informs different scenarios of flow regime variations: reservoir-related widening, morphological changes in river platforms, and hydrological signals (flow conditions: widening/high or narrowing/low) (Fig. S1, see Methods). Separating these different signals is critical to

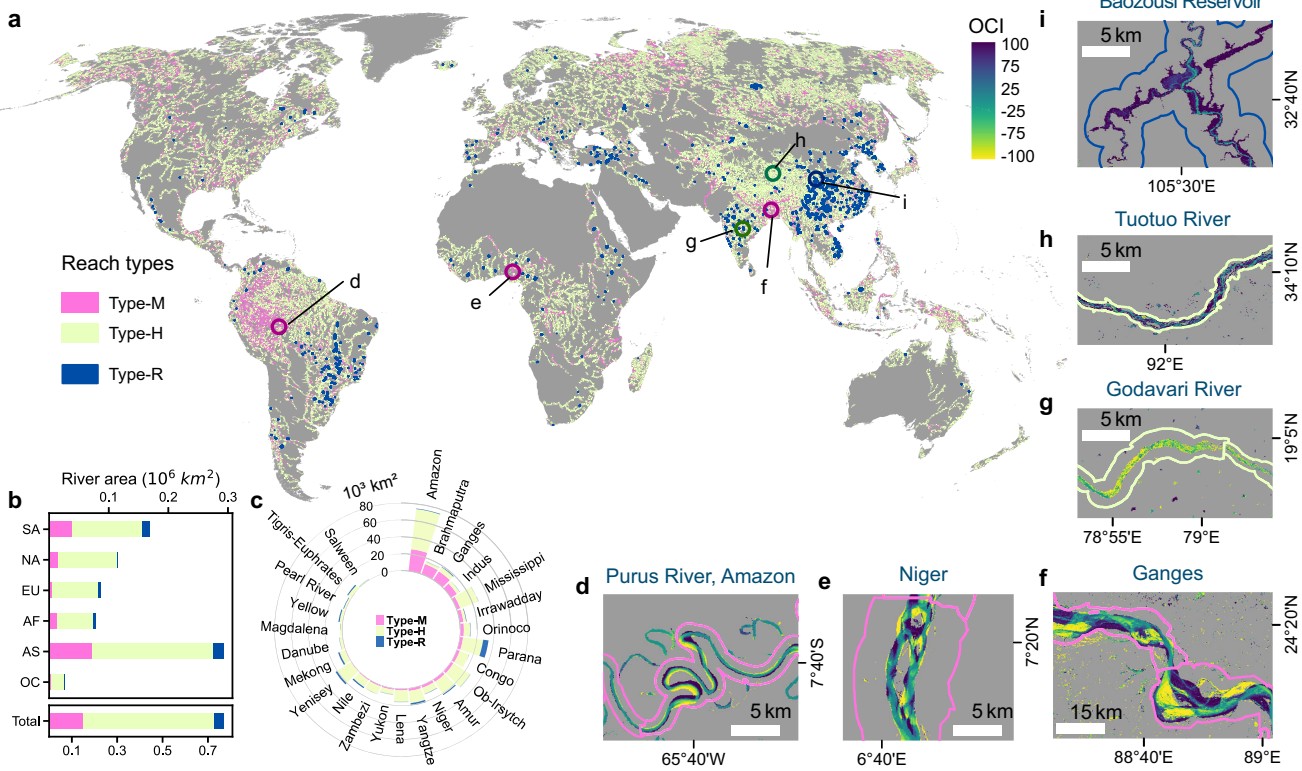

**Fig. 1 | The global distribution of different types of river extent changes in the early 21st century: morphological dynamics (Type-M), hydrological signals (Type-H), and new reservoir-type river reaches (Type-R). a** The global map of different types of changes. **b, c** The areal statistics of different reach types in six continents (NA North America, SA South America, EU Europe, AF Africa, AS Asia, and OC Oceania) and in the 25 mega basins. Zoom-in maps (**d–i**) exemplify different types of water extent changes on the Occurrence Change Intensity (OCI) map.

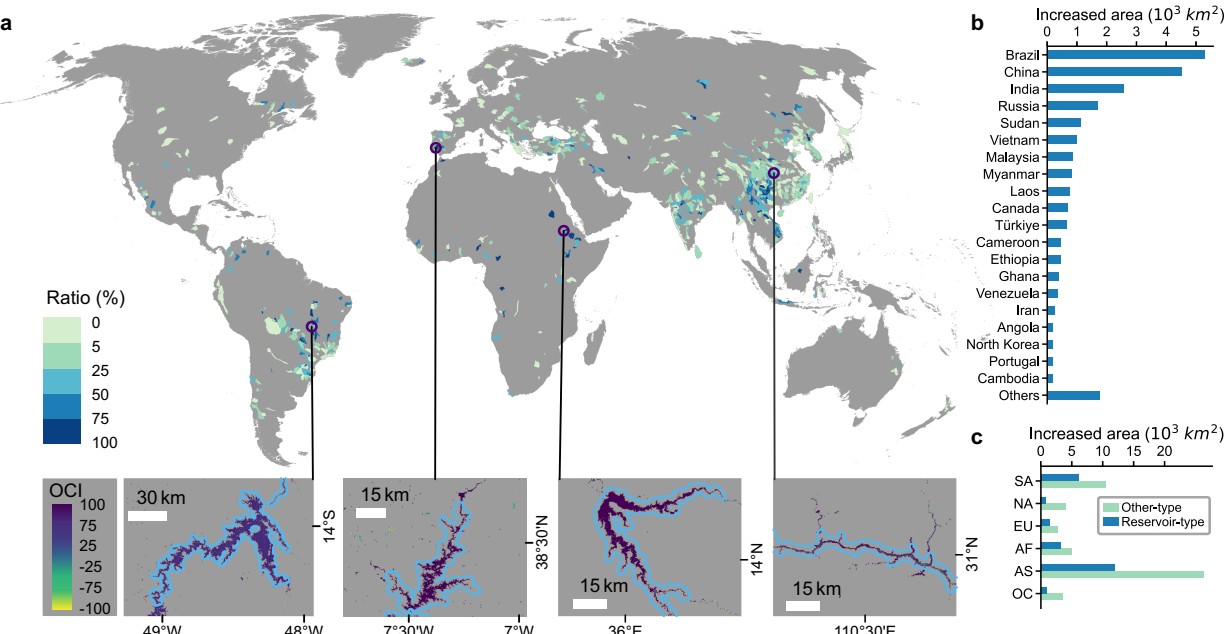

**Fig. 2 | River widening caused by new dams (Type-R). a** The ratio of expanded water extent (areas of significantly increased inundation frequency) to the basin-wide total river area reveals dam-related river widening across specific regions of the world. The bottom insets (from left to right) are four typical giant dams of the world, the Serra de Mesa Dam (Brazil), the Alqueva Dam (Portugal), the Upper Atbara and Setit Dams Complex (Sudan), and the Three Gorges Dam (China). **b** Statistics of the total expanded river area due to new dams in counties/regions. **c** Comparison of the increased river area due to dam construction with other types of expansions (Type-H) in each continent.

understanding river behaviors in the past two decades. To our knowledge, we provide a first-ever attribution of different types of river extent changes on a global scale. For this, a new reservoir inventory is compiled to define new reservoir-type river reaches (Type-R), and a machine learning classification approach is employed to classify the remaining basin-wide river changes into morphological dynamics (Type-M) and hydrological signals (Type-H). The machine learning model is founded on an expert-based interpretation guideline based on frequency change statistics, morphological attributes, and typical styles interpreted from the OCI map (Figs. S3, S4, see Methods). We aggregate the results from the level-9 basin units in the HydroBASINS dataset[30], to broader spatial scales, including level-6 basins, 25 mega river basins, and continents (Fig. S5). Focusing on the Type-H areas, we identify hotspots where rivers present dominant signals of widening or narrowing and discuss potential driving forces, including climate change and/or human interventions on the basis of long-term climatic records[31–35], satellite-based indicators of social activity intensities, human density, and published reports and literature.

## Results
### Morphological dynamics of rivers
We investigated water extent changes on 769,391 km² of river area (referring to Methods for the definition of river area) with a total length of 2,097,799 km in 94,659 level-9 basins. Our classification results reveal about one-fifth (19.6% by area) of the global rivers show active morphological dynamics (Type-M in Fig. 1). On this type of river basin, high percentages of narrowing and widening are observed along different banks/locations of river reaches, which are associated with the variations of flow regimes of meandering, braided, anabranching or wandering river channels (exemplified in zoom-in maps in Fig. 1). These river basins concentrate in the world's largest river systems, including five originating from the High Mountain Asia (Brahmaputra, Ganges, Indus, Irrawaddy, Amu Darya), the upper Amazon of South

America, the Yukon and the lower Mississippi of North America, the middle-lower Niger and the middle Congo of Africa, and the northern Dvina of Europe (Fig. 1, see detailed maps in Fig. S6). About one-quarter of the Type-M river areas are respectively distributed in the upper Amazon and rivers originated from the High Mountain Asia (Fig. 1c), where the highest percentage of morphological dynamics (40–80%) among the 25 mega basins were observed (Fig. 1c, Fig. S6). The morphological changes occur in certain geometric (high sinuosity/multi-tread), hydrological, and geological settings (slope, stream power, bank material, erosion, and sedimentation), which reflect decreased channel stability or active river channel evolution[29,36]. A recent study[37] finds that the prevalence of anabranching rivers is associated with low water surface slopes, wide floodplains, unconsolidated sedimentary substrates, and net sediment storage. Particularly, rivers originating from the High Mountain Asia are characterized by large elevation drops in the upstream areas and high sediment flux, which may exert forces on the formation and migration of meandering or braided river channels in the middle and lower reaches[38–42]. Historical satellite images reveal that lower Brahmaputra experienced remarkable channel migration during 1984–2016 (Fig. S7), which is probably attributed to low hydraulic efficiency partly due to high sediment delivery, tectonic activities, and intense erosion[39,43].

Besides natural forcings of evolution, other factors, including climate change and human interventions (e.g., water impoundment or transfer projects), could also generate impacts on altering river morphology[5,44,45]. For example, studies have shown that altered runoff from monsoonal rainfall and snow and ice meltwater may somewhat contribute to the increased number of channels of the Ganges between 1980 and 2005[41], and river channel shifting in the downstream areas[38,40]. However, identifying the causes of morphological dynamics on a global scale is challenging. Combined on-site and laboratory observations with numerical modeling approaches are needed to understand the process of river morphological changes in response to natural and external forcings.

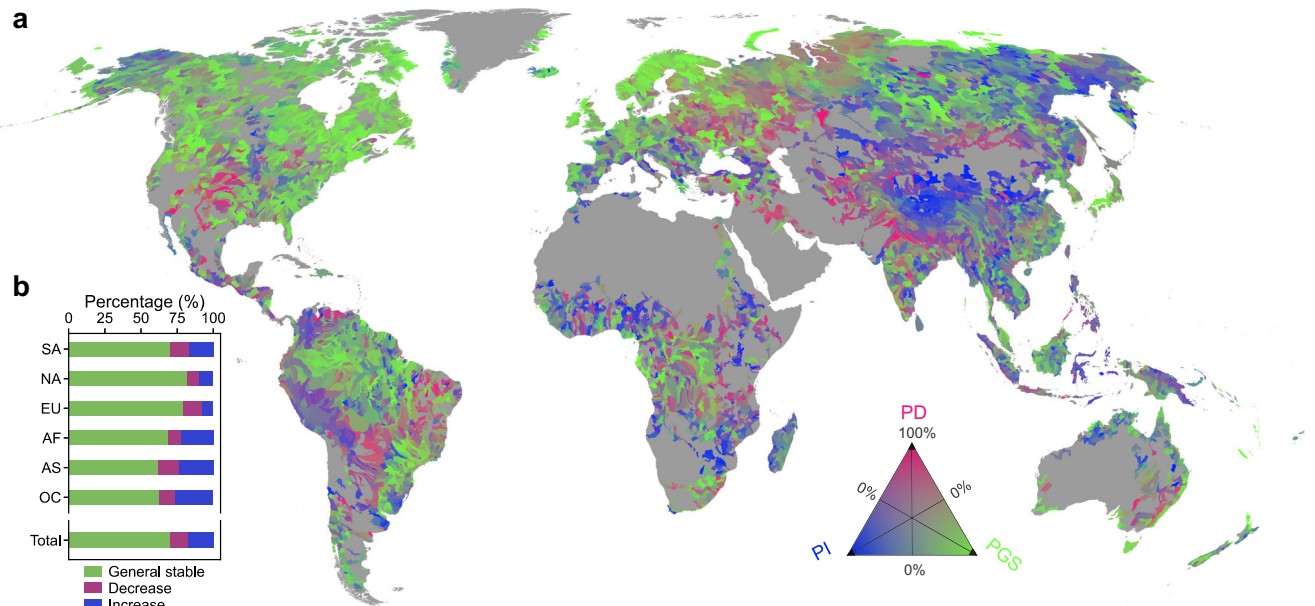

**Fig. 3 | Water extent changes from 1984–1999 to 2000–2018 on Type-H rivers (excluding Type-M (morphological dynamics) and Type-R (new reservoir-type rivers). a** The global map of water extent changes composited from the percentage of increase (PI, blue), the percentage of decrease (PD, red), and the percentage of generally stable (PGS, green) at the level-6 basin scale, with transparency of each color ranging from 0% to 100%. **b** The percentages of different classes (PGS, PD, and PI) aggregated in each continent.

## Global river widening due to new dams

Dam construction represents human interventions that directly lead to significant river flow regime expansion along the upper river reaches. Our reservoir inventory includes a total of 1118 newly-dammed reservoirs on the SWORD river networks (see Methods and Fig. S8). These reservoirs, located in 732 level-6 hydro-basins, cover a total of 40,757 km² water area (5.3% of the global river area), 60.0% of which are expanded flow regimes (areas from nearly no water flow to constant flow). This expanded water area amounts to 23.4% of the total river area in these hydro-basins, equaling river area expansion by 30.5% during the two epochs. The ratio of dam-related water expansion to the basin-wide total river area quantifies the extent of the dam widening effect (Fig. 2), which is spatially diverse and remarkably prominent in the developing countries in Asia (e.g., China, India, Vietnam, Malaysia, Myanmar, and Laos), South America (e.g., Brazil), and eastern and western Africa (e.g., Sudan, Ethiopia). Brazil, China, and India are the top three countries with the largest dam-related water area expansion, which respectively contribute to 21.7%, 18.5%, and 10.5% of the global dam-related increased water area (Fig. 2b). Dam-related river expansion was also remarkable in eastern Canada, Türkiye and Portugal in the southern and western Europe.

Globally, dam constructions emerged as a major contributor to the river widening signal, with the increased river area accounting for 31.9% of the global increased river area (excluding basins of morphological dynamics). This ratio (Fig. 2c) is exceptionally high in Africa (38.7%), South America (36.9%), and Europe (34.3%), where river widening is weaker than that in Asia (see discussion next). The contribution of dam-related water expansion to the basin-wide increased area mostly varied between 50% and 95% (Fig. S9), demonstrating a strong signal of dam widening at local scales. We exemplify river widening at four of the world's largest dams (Fig. 2), Serra de Mesa Dam on Tocantins River (Brazil), Alqueva Dam on Guadiana River (Portugal), Upper Atbara and Setit Dams Complex on Setit (Sudan), and The Three Gorges Dam on the Yangtze (China), which expanded the total river area in their basins by 32.9%, 756%, 503%, and 47.1% (contributed to 99.2%, 95.5%, 96.9%, and 87.8% to the total basin-wide river expansion), respectively.

## River flow regime changes in response to climate and anthropogenic forcings

By excluding basins dominated by morphological dynamics and new reservoir-type river reaches, here we focus on mean flow regime changes in the remaining 578,035 km² river area in 8863 level-6 basins. Globally the areal percentage of the significant increase (moderate increase) reaches 9.0% (8.6%), higher than that of significant decrease (moderate decrease) at 4.8% (7.4%). Meanwhile, 70.2% of the river areas are relatively stable in water extent. These global statistics indicate channel widening or increasing river flow as prominent characteristics during the past decades. Compositing percentages of the pixels in the three directions (Increase, Decrease, and Stable) in each level-6 basin reveals contrasting patterns of river changes (Fig. 3). Basins dominated by the 'Increase' signal (Fig. 3, in blue) mainly concentrated in the Tibetan Plateau, central and eastern Siberia, Southeast Asia, coastal west and southeastern Africa, and western South America, whereas the 'Decrease' signal (Fig. 3, in red) are more scattered in space and relatively strong in the India subcontinent, central Eurasia, the southern Great Plains of the United States, and central South America. The number of basins dominated by the increase and decrease signal accounts for 14.7% (n = 1300) and 6.0% (n = 531) of all examined basins, respectively. River extent tended to be generally stable in developed regions in North America, northern and western Europe, and some remote regions, such as the Amazon River and the Congo River basins (Fig. 3, in green). Around 60% of the examined basins (n = 5290) show dominated signal of stable water extents (with the percentage of general stable class larger than 50%).

The contrasting pattern of river extent changes can be exemplified in the 25 mega river basins (see location in Fig. S4). The Yellow, Brahmaputra, and Irrawaddy are the top three basins with the highest percentages of increased flow area in Eurasia (65.6%, 47.5%, and 33.9%, respectively), while Magdalena (50.1%), Niger (32.0%), and Yukon (16.6%) rank the highest percentage in South America, Africa, and North America, respectively (Fig. 4a, Table S1). The top four basins (Indus, Ganges, Brahmaputra, and Irrawaddy) all in Eurasia show as high as 21–29% of extents with narrowed water flow (Fig. 4b). In terms of net increase, which refers to the difference between the increased

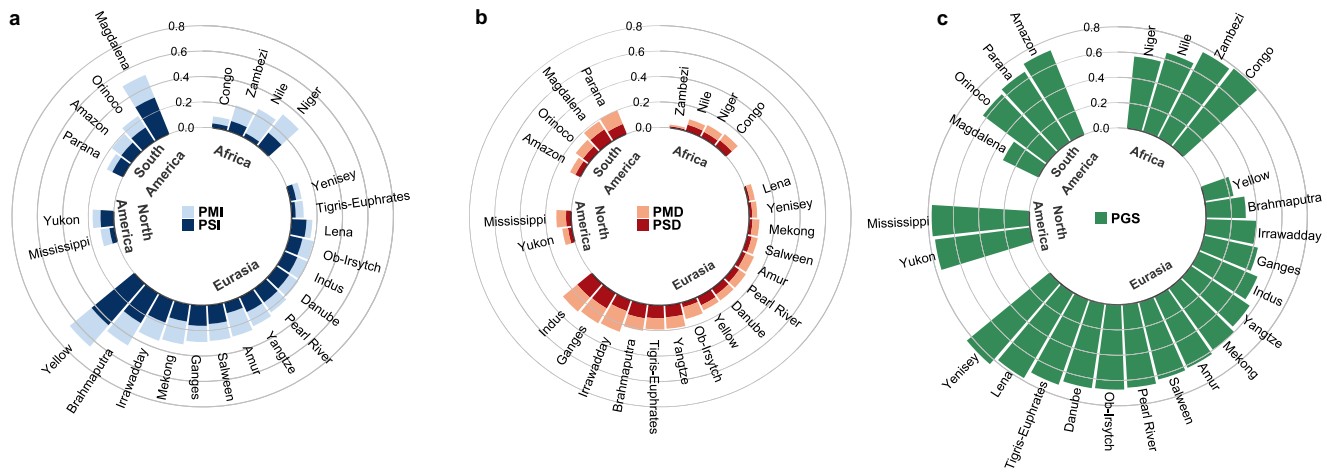

**Fig. 4 | Water extent changes in 25 mega river basins. a** the percentage of increase (PI). **b** the percentage of decrease (PD). **c** the percentage of general stable river areas (PGS).

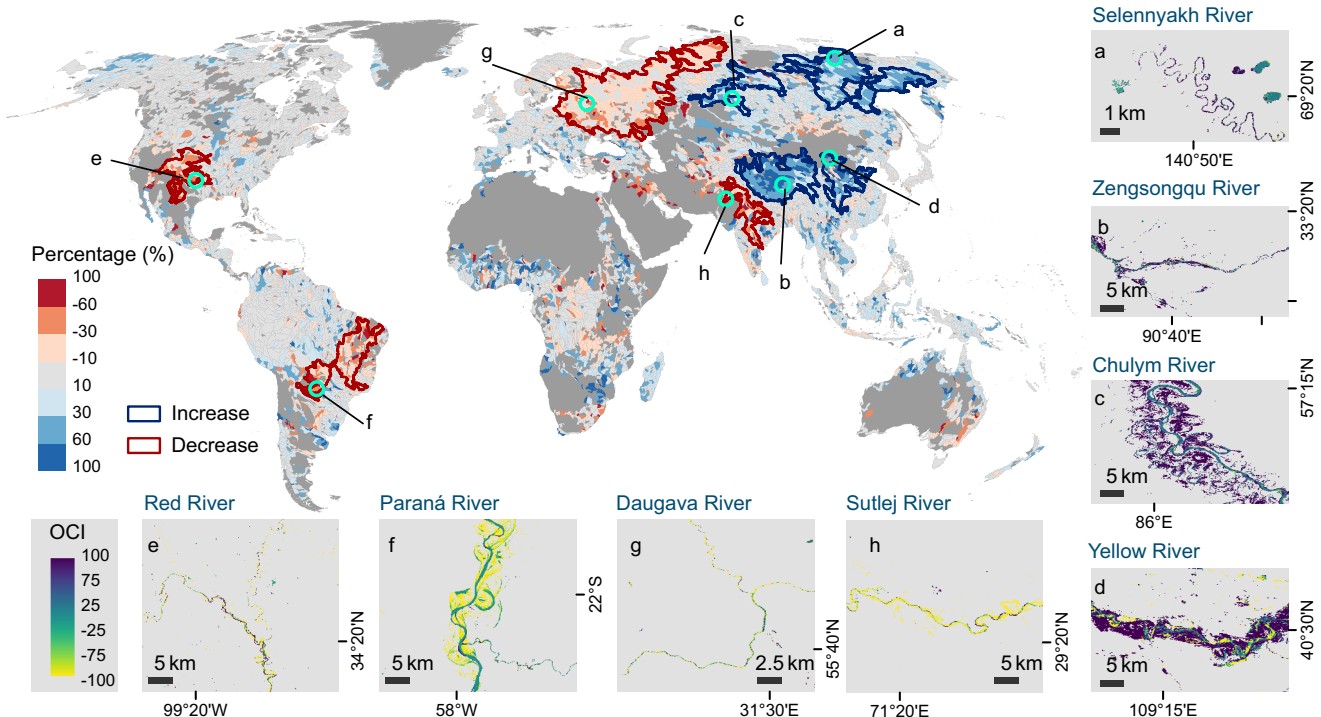

**Fig. 5 | Hotspots of river extent changes.** The upper map shows the top four largest hotspots with positive net increase (**a–d** in blue outlines) and those with negative net decrease (**e–h** red outlines), overlaid on the map showing the basin-wide relative magnitude of net increase (the areal difference between increase and decrease divided by the total river area). The green circles on the map indicate the locations of the zoom-in maps within the labeled hotspot region.

and decreased proportions, the majority of the mega basins (20 out of the 25) are positive, with increases greater than decreases. The top three basins with the highest percentage of increase (Yellow, Magdalena, and Brahmaputra) are also the top three basins with the highest net increase, while the Tigris–Euphrates in Middle East, the Indus in northern Indian subcontinent, and the Parana in northern South America show the highest net decrease (−5-−10%), signifying the strongest signal of river narrowing. In contrast to highly dynamic changes in river basins in South and East Asia, stability is the dominant signal in the middle regions of these continents/subcontinents, such as middle North Asia (Yenisey, Lena), middle North America (Mississippi), and Middle Africa (Congo, Zambezi), all with percentages of stability over 75% (Fig. 4c).

We retrieved hotspots of significant river changes by using the metrics of the relative magnitude of net increase (the areal difference between increase and decrease divided by the total river area). The largest four hotspots with a high percentage of net increase (denoted as positive hotspots) are all observed in Asia, including eastern Siberia (a), Tibetan Plateau (b), middle northern Siberia (c), and middle eastern Asia (d), while the largest four hotspots of decrease signal (denoted as negative hotspots) consist of the Great Plains in central North America (e), middle-eastern South America (f), western Siberia (g), and northern India (h) (Fig. 5). Details of river narrowing and widening in these hotspots are shown in insets in Fig. 5. Except the middle East Asia (mostly the Yellow River basin), the positive hotspots belong to high-latitude (a, c) or high-altitude (b) cold climate. In contrast, the negative

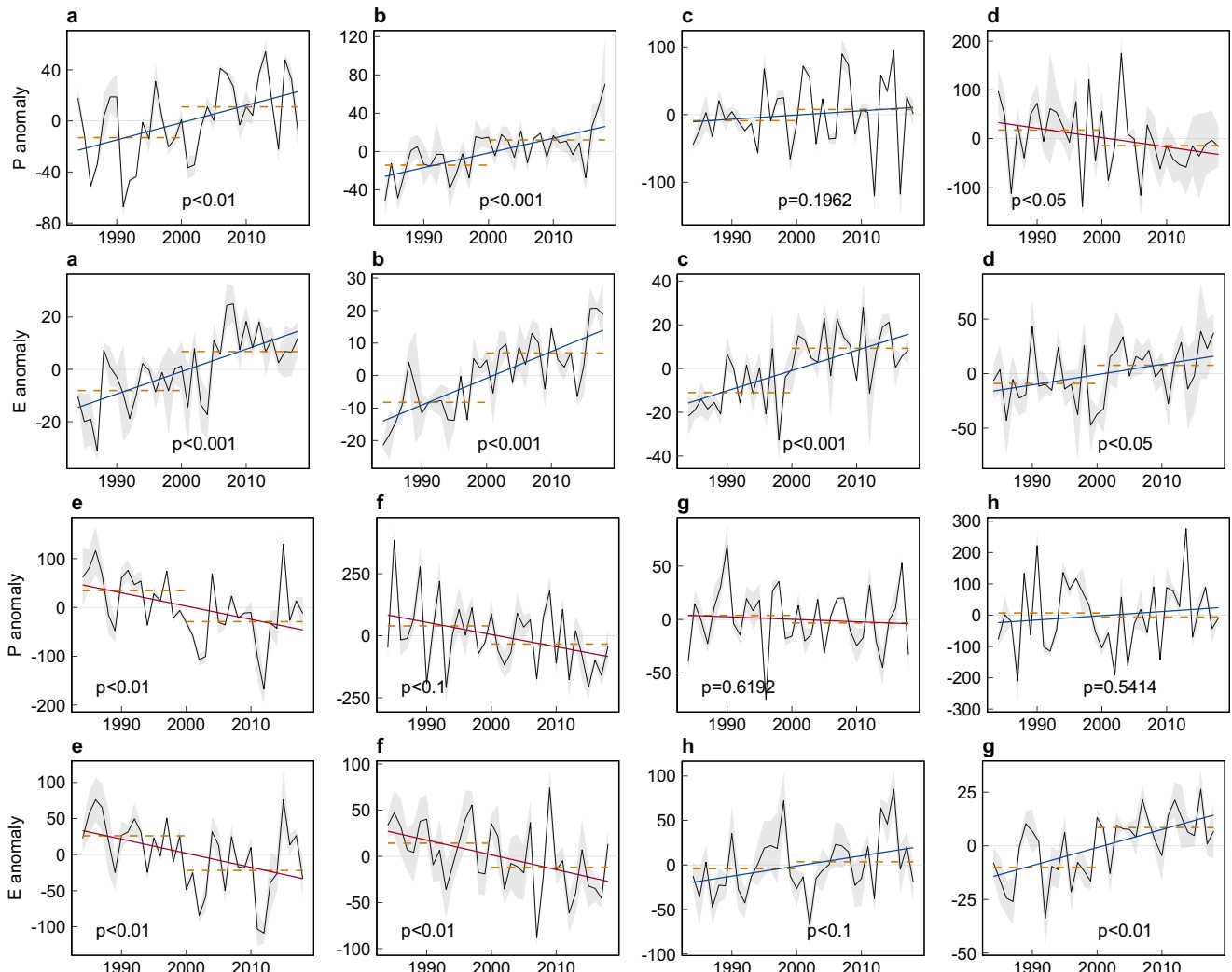

**Fig. 6 | The time series of precipitation (P) and evapotranspiration (E) anomaly for the eight river change hotspots.** See locations of the hotspots (labeled as **a**–**h**) in Fig. 5. The solid red lines show the fitted linear trend (*p*-value for the trend significance) of the time series. The dashed lines represent averages in the first epoch (1984–1999) and the latest epoch (2000–2018). The P and E data are averaged from different sources of datasets (see Methods), with the gray shades representing uncertainty according to the inter-model standard deviation.

hotspots are located in the continental interior characterized by arid or semi-arid climates. According to the global aridity map[46,47] (Fig. S10a), the three negative hotspots (e, f, h) show a higher level of aridity (lower aridity index) than that of the positive hotspots except for (b), which is typical of cold and arid high-land climate (Fig. S10b).

Climatic datasets and published materials reveal that climatic and anthropogenic factors played different roles in causing the river flow changes in these hotspots. Climate change, typically changes in precipitation and evapotranspiration, largely determines regional water balance and river flow variations. The climatic datasets from different sources (see Methods) reveal a significantly higher level of yearly precipitation in the latest epoch (2000–2018) than that in the previous epoch (1984–1999) in the pan-Arctic (a, c) and the Third Pole regions (b) (Fig. 6). For these regions, the more humid climate during the first two decades of the 21st century has been supported in previous research[48–50]. In addition to the precipitation increase, intensified glacier melting or permafrost thawing in response to warming climate[51–54] may also have promoted rising river flow in these regions characterized by high coverage of permafrost and/or glaciers[55–57]. In summary, climate warming and wetting trend are likely the main cause of the river widening signal on the pan-Arctic and third-pole regions where human impact has been limited.

In contrast to the wetting trend in the cold regions of Asia, decreasing precipitation trends are observed in the two negative hotspots (e, f), and significantly increasing trends of actual evapotranspiration are observed in the other two (g, h). These changes imply drying climatic conditions during the latest epoch, which could lead to significant river narrowing in these areas with delicate water balances. For instance, the Colorado River in North America has suffered a decrease of 9.3% in annual mean discharge per Celsius degree of warming[58]. Eastern Brazil has experienced severe drought[59,60], leading to the terrestrial water storage declining at 16.7 ± 2.9 Gt/yr during 2002–2016[21]. Approximately two-thirds of the global endorheic water loss stems from central Eurasia, covering one-third of the endorheic land-mass[61,62].

The above analysis supports that climate change generally explains the contrasting pattern of river extent changes in these eight hotspots, except hotspot (d) where no significant wetting trend was observed. The pattern of climate change is evident from the map of the decadal changes of the two climatic variables (Figs. S11, S12), which reveal generally consistent patterns of wetting/drying climate, despite uncertainties in the quantities and scales due to inherent limitations in the climatic observation or modeling techniques (Fig. S13). In addition to climate change, we must stress clear human influences on the global

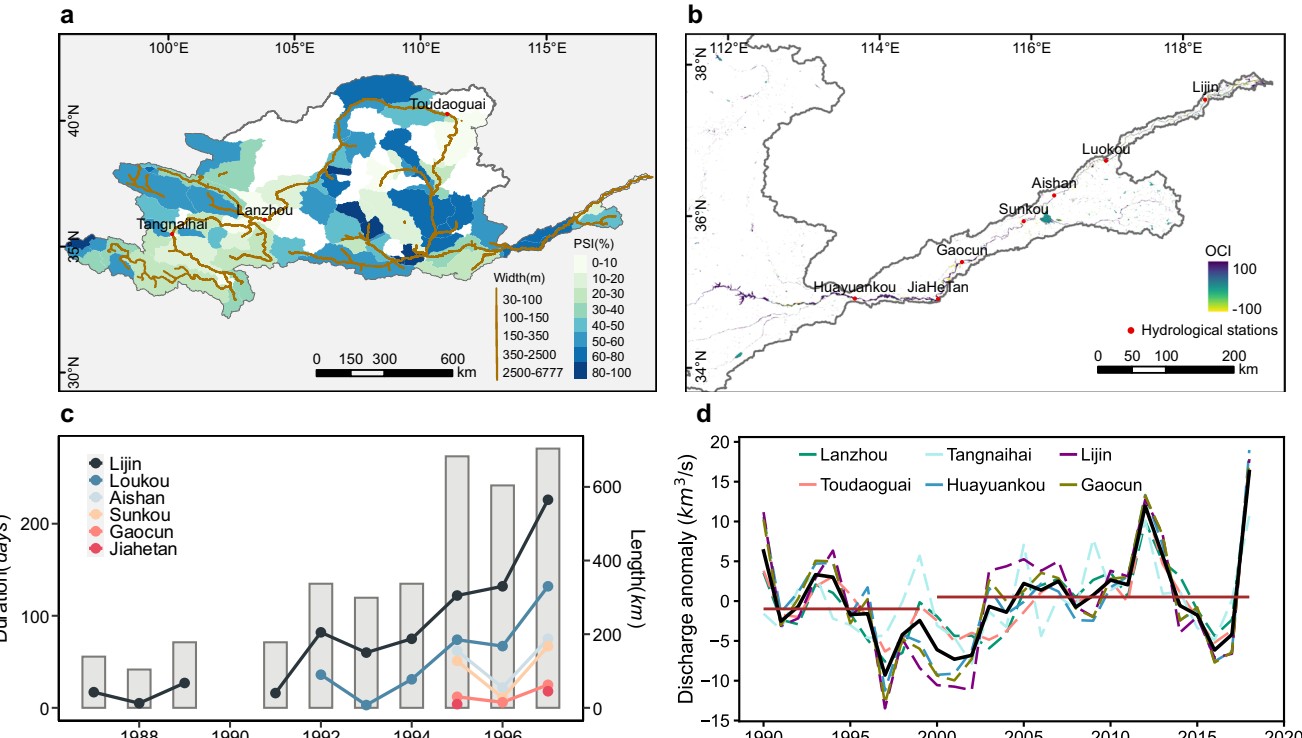

**Fig. 7 | Flow cutoff and recovery in the Yellow River. a** The percentage of significant increase class in the Yellow River basin. **b** The OCI map on the lower reaches. The distribution of gauging stations is shown in red dots in the maps of **a**, **b**. **c** The dried-up distance and duration (no available data in 1984–1986 and 1990) in the Yellow River. **d** The long-term discharge anomalies measured at six gauging stations along the river, with mean anomalies shown in solid black line. The solid brown line denotes the mean discharge anomalies during 1990–1999 and 2000–2018.

river extent changes. Besides the remarkable dam-related river widening we discussed before, the human impact on river changes is two-fold, according to current observations. First, human interventions amplified the regional water deficit in the four negative hotspots of arid or semi-arid regions, leading to further river narrowing through large-scale irrigation and extensive groundwater abstraction to satisfy the increasing water demand. This is particularly true for what occurred in eastern Europe and northern India[21], where groundwater abstraction has aggravated drought and soil-moisture deficits, impacting regional water cycle[63–65] and terrestrial water storage[66,67]. In Oklahoma in central North America, high groundwater withdrawal for irrigation and domestic usage caused a severe decline in groundwater discharge and stream leakage, which was the major contributor to the declined streamflow between 2001 and 2014[68]. These studies infer clear human influences on the narrowed river extents in arid- and semi-arid regions.

Another form of human impact on river changes lies in the river flow recovery in the Yellow River basin (the core regions of hotspot d). The high percentage of increased flow coverage in this region is essentially a recovery from the no-flow cutoff (less than 1 m³/s runoff at the Lijin hydrological station situated at the lowest reach of the Yellow River). According to earlier literature[69–72] and hydrological records (Fig. 7), the Yellow River experienced multiple no-flow cutoffs in the lower reaches and the headwaters during the last decades of the 20th century. The dried-up magnitude and duration reached a maximum in 1997 (around 700 km and 226 days)[70,73], due to excessive water consumption, the absence of an effective and integrated institutional framework for sustainable water use, poor dispatching-storage and anti-dispatching abilities, and a drying climate[69,73,74]. However, the dried-up situation of the Yellow River gradually improved during the early 21st century, with the implementation of the Yellow River Unified Schedule and a series of water conservation techniques to optimize water supply for different needs, including maintaining ecological river flow[70,71,75]. The recovery of the river flow can be revealed from the increasing trends of discharge recorded at gauging stations (Fig. 7) and, consequently, a significant increase in water flow extents, particularly along the lower reaches. The case of the Yellow River demonstrates a typical example that without a concurrent precipitation increase, a river recovers from extreme decreases in flow regimes through improved water-use policy and techniques.

## Stable river extent in developed regions

Over half of the global river area shows relative stability in flowing extents. The highest percentage of river stability is observed in North America (82.1%), followed by Europe (79.5%) and South America (70.5%) (Fig. 3). River channels in developed regions, such as counties in northwestern Europe (e.g., Finland, Sweden) and North America (e.g., Canada, the United States), were more stable than those in the developing regions in Southeast Asia (e.g., Myanmar, China) and South America (e.g., Bolivia, Peru) (Fig. 8). This contrast suggests that the stability of river extent may be correlated with the socio-economic development level, which can be inferred from the nighttime light intensity map[46,76] and human population density map (Figs. S14, S15). In addition, we find a good spatial correlation between the basin-wide river channel stability and that of nighttime light intensity in the primary habitat of human beings; and the percentage of stable river areas generally increases with nighttime light intensity (Fig. 8). This phenomenon may be associated with the situation that built-up areas are often sited far from highly dynamic regions (e.g., steep land, headwaters, and highly active/unstable fans and floodplains). On the other hand, the highly developed river embankment projects including river levees implemented earlier (e.g., since the mid-20th century) in developed regions can help confine the extent of river flow. For example, approximately 160,000 km of levees were constructed

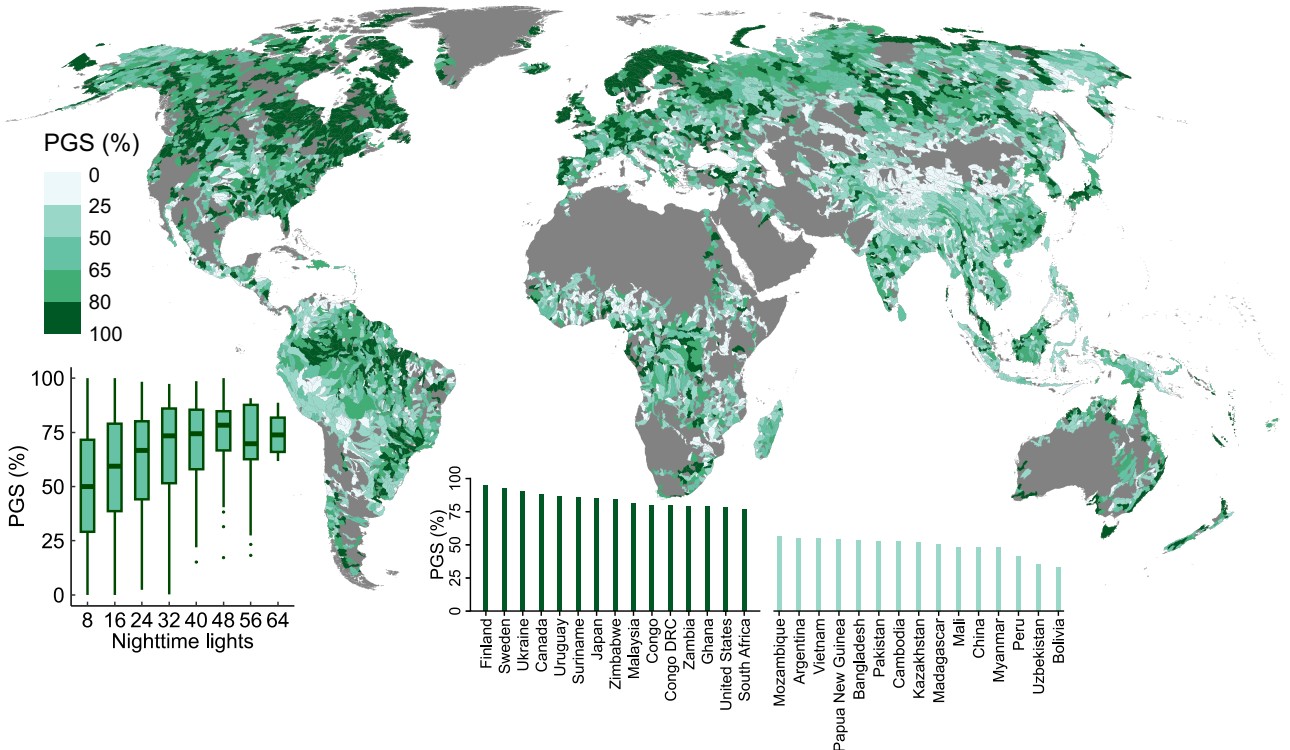

**Fig. 8 | The distribution of stable river extent and its correlation with nighttime light intensity.** The map shows the percentage of river area with generally stable extent (PGS, %) in each level-6 basin. The top 15 countries/regions with the highest PGS and the lowest PGS among the top 50 counties with the largest river area are shown on the bottom inset. The left inset shows the PGS statistics in relation to nighttime lights for each basin where the population density is larger than 1 person/km².

across the United States to protect urban areas and cropland from flooding[77,78]. The Mississippi levee system is the largest globally, stretching more than 5600 km (Fig. S16)[79]. Many countries across Europe, including the United Kingdom, France, Germany, Netherlands, and Poland, have several thousands of kilometers of river levees (England: 7100 km; France: 8100 km; Germany: 8000 km; The Netherlands: 1800 km; Poland: 6600 km)[80]. Such developed river embankment networks may have reduced the disruption of flood pulse and bank erosions, thus contributing to stable river flow regimes[81–84].

In summary, we present decadal river extent changes on the global river networks (a total of 769,391 km² by area and 2,097,799 km by length) through a synergistic usage of multi-themed, satellite-based observations. Our global-scale analysis reveals the patterns of world's river flow regime changes in the early 21st century and its attribution to morphological dynamics, climate change, and anthropogenic stressors. We reveal that river morphological dynamics, such as channel shifting and anabranching, prevailed in about 20% of the global river area between the epochs of 1984–1999 and 2000–2018. About 33% of the remaining river area experienced significant regime changes, of which two-thirds exhibited river widening and one-third narrowing. Booming reservoir constructions, mostly in developing regions in Asia and South America, contributed to ~32% of the river widening. In contrast to the relatively high stability of river width in developed regions, we observed intensive river narrowing in some arid endorheic basins, raising concerns about exacerbated water deficit caused by climate change and rising water and energy consumption demands. Whereas in the remote and alpine regions, climate forcing, including contrasting patterns of precipitation and evapotranspiration and cryospheric responses (increases of snow/ice meltwater), has likely been responsible for the prominent river channel widening, especially in the Tibetan Plateau and eastern Siberia. The holistic map of river

extent changes worldwide highlights contrasting river changing patterns and drivers between arid and humid regions and between developed and developing regions. River widths maintain relatively high stability in developed regions, partially attributable to the effect of river infrastructure. Our analysis provides a global-scale but spatially-explicit guidance for better prioritizing future river protection and restoration efforts under the UN 2030 Agenda for Sustainable Development, which calls on international actions to track the spatial extent and condition of water-related ecosystems.

## Methods
See Fig. S2 for a summary of the data and methods used in this study.

### Statistical variables and the indication of river extent changes
The OCI map was created by averaging all surface water occurrence differences derived from homologous pairs of months between the two epochs (from 16 March 1984 to 31 December 1999 and from 1 January 2000 to 31 December 2018) from the GSW database[23]. The OCI values, ranging from −100 to 100, indicate different directions and magnitudes of water cover frequency changes, which are categorized into five classes: Significant increase (SI, $100 \geq OCI \geq 75$), Moderate increase (MI, $75 > OCI \geq 25$), Generally stable (GS, $25 > OCI > -25$), Moderate decrease (MD, $-25 \geq OCI > -75$), and Significant decrease (SD, $-75 \geq OCI \geq -100$). The proportions of pixels in the increase (SI + MI), decrease (MD + SD), and general stable class (GS) were used to indicate river extent changes in broadly three directions. The SI and SD classes reflect high confidence in changes in water extents and flow conditions, while the MI and MD classes bear a relatively lower magnitude/confidence.

We illustrated representative scenarios of inundation frequency changes along river channels (Fig. S1) that are associated with river

morphological dynamics and hydrological signals. River morphological dynamics reflect relocation of flow regimes of meandering, braided, anabraching or wandering river channels. Other types of river change signals include widened rivers in the upstream reaches due to the construction of new dams, rising/declining runoff and correspondingly widened/narrowed river channels due to wetting/drying climate, or changes in water supplies from the cryosphere. The relocation, widening, and narrowing of rivers will essentially lead to changes in water flow extents which can be examined from the OCI map, such as clusters of highly positive (SI/MI) or negative (SD/MD) pixels or both. River embankments and levees constructed for flood prevention will generally lead to confined river flow extents and stable widths, and thus OCI values will approximate zero (GS) in these areas.

We relied on the statistics of the OCI values in different classes to infer the type and the pattern of the basin-wide river change signal. On the OCI map, land surfaces that are never mapped as water, or not covered by Landsat imagery (such as most of the Polar regions), or no homologous months for inundation frequency change comparison, were assigned with filling values, and they were all masked out in the statistics. Therefore, the river area in this study refers to the total number of valid pixels within the river buffer extents multiplied by the per pixel area (30 m by 30 m). The OCI map and river buffer extents were all projected to equal-area projections for different continents.

## Statistical extent of rivers

To retrieve river extents, we implemented a buffering method that only included pixels historically inundated by rivers and masked out by a global lake inventory. We used the version SWORD v2 (https://zenodo.org/record/3898570#.YyE8qqRBwuW) as the primary reference for global river networks and river morphological attributes. SWORD represents one congruent product of river networks consisting of river reaches (~10 km long) and nodes (~200 m spacing) with important hydrologic and morphological attributes (width, slope, meander length, sinuosity, and number of channels), generated by combining several global river- and satellite-related data sets. River networks and width attributes in SWORD were inherited primarily from the Global River Widths from Landsat (GRWL) database (refer to Altenau et al.[28] for more details about SWORD). To derive potential river water extent, we employed the seamless level-9 basins of global coverage from the HydroBASINS database[30] as the basic scale of river segment division. River centerlines in SWORD were clipped by each level-9 basin polygon to form individual basin-scale river networks. To avoid sliver river centerlines and consequently unreasonably small statistical area for each basin, we dissolved short river segments to the closest nearby basin by basin code until the river centerlines in each basin are longer than the mean meander length. This results in 88,162 level-9 basin units covering SWORD, 35.5% of which are merged level-9 basins from HydroBASINS. These basin-scale river centerlines were masked by a global lake inventory retrieved from the HydroLAKES database containing 1.4 million lake polygons[85]. We referred to the attributes of lake type, lake area, and shoreline definition in the HydroLAKES database in defining the lake inventory, and particularly kept eight super-large lake-type reservoirs (Table S2) as reservoirs rather than lakes, considering that changes around these lake-type reservoirs represent changes from rivers to reservoirs.

The attributes of individual basin-scale river reach (including mean width, mean slope, mean number of channels, mean sinuosity) were calculated by averaging the corresponding attributes of nodes that intersect with basin river centerlines. Then buffer zone for each basin centerline was created with a buffer distance triple the mean river width for multi-thread or sinuous rivers (mean braiding index >1 or mean sinuosity >1.2, referring to thresholds suggested by Leopold & Gordon[29]) and twice the others. The river buffer zone was further clipped by the basin boundary to restrict the river extent statistics within the basin.

To complement the relatively sparse river networks in SWORD in High Mountain Asia, particularly in the inner Tibetan Plateau, we constructed an additional river vector dataset for this area based on the Maximum Water Extent data in the GSW dataset (Fig. S17). We manually eliminated non-river water bodies in the river vector dataset for this area. This data set supplements an additional 15729.1 km$^2$ of river area and 6497 new level-9 basin units.

## Identification of new reservoir-type (Type-R) river reaches

To identify river reaches impacted by the post-2000 reservoirs, we compiled a new global reservoir inventory by supplementing newly mapped post-2000 reservoirs to the published GeoDAR reservoir dataset compiled by various sources of information[86]. Our method of mapping new reservoirs consists of detecting new permanent inundation areas by segmentation of OCI maps (with a threshold of 85% according to our experimentation), masking out lakes with the HydroLAKES data, manual inspection against high-resolution satellite imagery in Google Earth to confirm the locations of new reservoirs, and retrieval of the boundary according to the maximum water extent product of the GSW dataset[22]. This method is similar to our recent publication[87]. By merging the two, we derived 111,8 post-2000 reservoirs intersected by the SWORD river centerlines. Among the 111,8 reservoirs, 562 were sourced from the GeoDAR. Reservoir boundaries in GeoDAR may not represent the maximum reservoir water areas. Therefore, we buffered their boundaries with a distance of 100–1500 m (proportional to the reservoir size) to get the statistical extent. By erasing the river buffers with the maximum extent of new reservoirs, we separated the new reservoir-type river reaches from other river areas, enabling direct quantification of the contribution of new reservoirs to river changes at different spatial scales.

## Identification of morphological signal with machine learning models

For proper attribution of river changes, we identified basins where frequency changes are primarily related to morphological dynamics (such as meandering and migration) by testing three widely used supervised machine learning classifiers: eXtreme Gradient Boosting tree (XG-Boost)[88], Random Forest (RF)[89], and Feedforward Neural Network (FNN)[90]. For the supervised classification, we randomly sampled 10% (9249) of the level-9 river basin units (with new reservoir-type river reaches masked out) for manual interpretation of the signal type (Type M: morphological; Type H: hydrological). Our basin-by-basin interpretation refers to quantitative statistics of frequency changes and morphological features (sinuosity, number of channels), as well as typical examples of morphological changes illustrated on the OCI map, as outlined in a guidance map (Fig. S3). The underlying rationality is that if the river reach showed active morphological dynamics, we expect an obvious proportion of change signals and that the percentage of increase and decrease signals is comparable within the river basin (no dominant increase or decrease signal). Other references include that the river reaches possibly show high sinuosity or a large number of channels on average. From the OCI map, different styles of morphological changes can be clued, including interlaced increase and decrease signals along sinuous channels, flow regime dynamics of multi-thread/braided rivers, and channel shifting of anabranching/wandering channels. The interpretation results are cross-checked by participants and discussed for indistinct cases.

The manually labeled datasets (Fig. S4) consisting of 1948 Type-M basins and 7301 Type-H basins were set as input for the training machine learning models for binary classification of all basins. For model training, we used 80% of the labeled basins as the training set and the rest 20% for validation. For results with the best performance (the highest Kappa coefficient), each model was trained 500 times, with different ratios of Type-M and Type-H basins for training at each time. We tuned the parameters of the three modeling methods

according to the specifications in Brownlee[91]. We tested a variety of metrics and finally employed basin-wide morphological metrics (sinuosity, number of channels), and frequency change statistics (PSI, PMI, PGS, PMD, PSD, PI, PD, |PI − PD|, |PI − PD|/(PI + PD)) as input variables in the modeling. Among the three machine learning classifiers, the XG-boost method yields the highest predicting accuracy (Table S3) and therefore was chosen as the final method. We also tested two schemes of machine learning, the global modeling and separate modeling for each continent, the latter yielding better results than the first according to the evaluation metrics (Table S4). The reason may be associated with different types of morphological signals for each continent and thus more adapted fitting of the classifiers. The classification method we employed achieves an overall accuracy of 94.55% (Kappa coefficient: 83.59%), which outperforms the global modeling methods (92.23%, 91.93%, and 89.99% for XG-Boost, RF, and FNN, respectively).

## Aggregation of statistics to broader spatial scales

We chose level-6 basins as the appropriate spatial scale of analysis, considering the compromise between spatial details and uncertainties associated with observation gaps in the SWORD and the OCI map. In this study, river areas cover 9171 level-6 basins (Fig. S4). For each level-6 river basin, we aggregated the statistics of total areas in the three directions of OCI changes (Increase, Decrease, and General stable) and for different types (Type-M, Type-H, and Type-R). The statistics allow separate evaluation of the basin-wide river changes associated with morphological dynamics, reservoirs and other types of signals. The level-6 basin-wide river changes were aggregated to selected 25 mega river basins in Eurasia, Africa, North America, and South America to briefly summarize global patterns of river flow regime changes. The term "mega" applied here largely depends on certain metrics, including drainage basin area, river length, and flow discharge, according to Best[5] and further filtering according to the total river area and the magnitude of river changes. We excluded seven basins, including the Mackenzie, the Columbia, and the St. Lawrence in North America due to their high percentages of stability, the Sao Francisco in South America owing to lack of sufficient observations, the Rhine and the Volga in Eurasia due to their relatively small drainage areas and stable river extents, and the Murry−Darling in Oceania because of its low mean annual flow.

## Hotspots of significant river change signals

By excluding all Type-M and Type-R areas, we assumed that the statistics of river extent changes are mainly indicative of hydrological signals driven by climate change, interdecadal climate variability, and/or indirect human impact. We used the relative magnitude of net increase, which refers to the difference between areas of increase and decrease of Type-H river area divided by the total river area, as a metric to indicate the dominance of expanded (widening) or shrink flow (narrowing), considering that different directions of frequency changes typically coexist in a single basin due to the high variability of rivers. The distribution of this variable follows a normal distribution with long tails, with the extremely high (low) values indicating the dominance of increased (decreased) river flow. On the basis of this variable map, we retrieved spatial clusters of basins dominated by high river flow (hot spots) and those dominated by low river flow (cold spots) at the confidence level of 90% (z-score value of 1.65) using the Hot Spot Analysis tool in ArcGIS (Fig. S18 for the distribution of all hotspots). We retrieved the outlines of the top four largest hot spots and cold spots for a detailed investigation of driving forces.

## Analysis of climatic variable changes for the hotspots

To understand the correlation between climate change and river extent, we analyzed changes in climatic variables for the positive and negative hotspots from 1984 to 2018 using multiple sources of precipitation and actual evapotranspiration products. Three kinds of precipitation data were used to derive the yearly time series of precipitation data: ERA5-Land (ERA5L) monthly reanalysis dataset at 0.1° × 0.1° (available at https://cds.climate.copernicus.eu/cdsapp#!/dataset/reanalysis-era5-land-monthly-means)[35], Climatic Research Unit gridded Time Series (CRU TS, v4.04) at 0.5° × 0.5° spatial resolution (available via https://crudata.uea.ac.uk/cru/data/hrg/)[32,92], and Multi-Source Weighted-Ensemble Precipitation (MSWEP, v2.8) dataset at 0.25° × 0.25° (available via http://www.gloh2o.org/mswep/)[34]. The evaporation data include ERA5L, TerraClimate-a high-spatial-resolution (1/24°, ~4-km) monthly climate and climatic water balance dataset (available via https://www.climatologylab.org/terraclimate.html)[33], and Global Land Evaporation Amsterdam Model (GLEAM, v3.6) dataset at 0.25° × 0.25° grid[31] (available via https://www.gleam.eu/). For each hotspot, we constructed the time series of annual precipitation and evaporation anomalies from these datasets by subtracting the multi-year (1984–2018) means from the monthly time series averaged on all grids within the region. Given uncertainties in the products, the multiple time series of anomalies were averaged to represent a merged trend, with the inter-model standard deviation as uncertainties. A non-parametric test technique (Mann-Kendall trend test) was used to assess the significance of the trend. To examine the spatial variability of changes in climatic variables, we mapped the decadal changes in annual precipitation and evaporation totals from the six products (Figs. S11, S12).

## Aridity and nighttime lights index

To measure the level of climate aridity globally, Zomer, Trabucco[47] developed the global aridity index by calculating the ratio of mean annual precipitation to mean annual potential evapotranspiration based on the WorldClim data[93]. The aridity index is larger in humid regions than in arid conditions[46]. We used the nighttime light data to evaluate the relationship between the intensities of human activities and the level of river stability. The nighttime light data were generated based on 30 arc-seconds cloud-free composited remote-sensing imagery from the Defense Meteorological Satellite Program Operational Linescan System. The data represent the average digital number values of cloud-free lights in visible bands. Details about the production of nighttime light data are given by Doll[76]. The percentage of nighttime light in a region reflects the economic prosperity and intensity of human activities. We excluded regions where nightlight intensity is not a good indicator of the degree of economic development in our statistics by referring to the population density map in 2000 from the Gridded Population of the World (v4) dataset[94] (Fig. S15). Specifically, basins with a mean population density less than 1 person/km² were excluded. The relationship between nighttime lights at the level-6 basin scale and inundation frequency changes was analyzed by computing the correlation between percentages of stable pixels in each level-6 basin to the basin-wide nighttime lights index.

## Uncertainties

The uncertainties of the OCI-based river extent change are associated with the robustness of the statistics algorithm and the spatial and temporal coverage of Landsat imagery. Here we discuss the uncertainties by comparing different sources of datasets including the correlation with in-situ discharge measurements at gauging stations. Compared to another Landsat-based dataset (GLAS), the OCI map shows similar frequency change patterns despite different algorithms (Supplementary Text and Fig. S19). This confirms the robustness of the Landsat-based river extent changes. Uncertainties mainly lie in the temporal unevenness of the Landsat observations, which tend to show the lowest coverage in the 1980s[22], implying that the frequency statistics in the first epoch may be more reflecting conditions in the 1990s for some regions (e.g., for the Northern Hemisphere high latitudes[22]). However, this temporal unevenness will probably not introduce much

bias in the GSW OCI map as the algorithm considers seasonal differences in comparing frequency[22]. To assess the seasonal asymmetry of observations during the two epochs, we produced a global map of the Pearson's correlation coefficient between the number of valid observations for each month of the first epoch and that of the latter epoch (Fig. S20). The correlation is relatively high across the vast majority of the global river basins (86% of river basins have an R-value greater than 0.5), indicating no significant seasonal bias on a global scale. We further evaluated the reliability of decadal river extent changes by referring to in-situ discharge observations derived from the Global Runoff Dataset Center (https://www.bafg.de/GRDC/EN/Home/homepage_node.html) and related literatures[95–97]. Our comparisons show that the basin-wide net changes of river extent (the difference between the increase and the decrease, PI − PD) are significantly correlated with the relative decadal difference of annual discharge at nearby gauging stations, which means the OCI metric is a generally reliable indicator of hydrological signal (Fig. S21).

## Data availability
The SWORD river database used in this study for organizing global river centerline and morphologic attributes is available at https://zenodo.org/record/3898570#.YzTsWHZBxZc. The river OCI data is from the European Commission's Joint Research Center for the Global Surface Water dataset and available at https://global-surface-water.appspot.com/. The data of river extent changes in Level-6 basin unit and the vector and attribute data of global rivers examined in this study are distributed through ScienceDB (DOI: 10.57760/sciencedb.07274). Other data in this study are available upon reasonable request to the corresponding author (C.Song, cqsong@niglas.ac.cn).

## Code availability
All analytical codes generated in this paper are available upon request (L. Ke, kelinghong@hhu.edu.cn, or C. Song, cqsong@niglas.ac.cn).

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

## Acknowledgements

We acknowledge the use of discharge gauging data from the World Meteorological Organization Global Runoff Data Center (GRDC) and the Ministry of Water Resources of China. This study was supported by the National Key R&D Program of China (Grant No. 2018YFD1100101, 2018YFD0900804, 2018YFA0605402, 2022YFF0711603), the Strategic Priority Research Program of the Chinese Academy of Sciences (Grant No. XDA23100102, XDA28020503), the Science and Technology Planning Project of NIGLAS (Grant No. 2022NIGLAS-CJH04, 2022NIGLAS-TJ18), the Second Tibetan Plateau Scientific Expedition and Research (STEP) (Grant No. 2019QZKK0202), and the National Natural Science Foundation of China (Grant No. 41901374; Grant No. 41971403; Grant No. 31922090). The authors would like to thank three anonymous reviewers for their constructive suggestions and comments that helped improve the manuscript.

## Author contributions

C.S., L.K., and Q.W. designed the study. L.K., Q.W., and C.S. performed data processing and research analyses on the river extent changes, with the help of J.W. in the mapping and validation of new reservoirs. C.S., Q.W., L.K., J.W., T.P., G.A., and B.Y. analyzed and interpreted the driving forces of global river changes. Y.S., X.D., Y.Z., J.W., L.W., K.L., T.C., W.Z., B.Y., and C.F. provided insightful feedback on methods and result interpretation. L.K., Q.W., and C.S. initiated the draft of the paper, with substantial contributions from all authors.

## Competing interests

The authors declare no competing interests.
