## [Peer Review File · Nature Communications]

Satellites reveal hotspots of global river extent changeREVIEWER COMMENTS

Reviewer #1 (Remarks to the Author):

The MS entitled "Satellite-observed global river channel changes driven by anthropogenic and climate forcings" submitted for publication in Nature Communications explores changes in current water extent along fluvial network worldwide between mid-1980s to 2018 and factors controlling them. If the descriptive approach is convincing and really interesting, the causal approach, still exploratory to assess the "main" drivers, is difficult to follow and not convincing at all, except the part on dams. Indicated drivers are mainly interpretations and not facts and it is very difficult to really assess what is true and not true, and to hierarch these potential drivers.

One of the main problem is linked to the fact that the explanatory variables are not clear (or introduced clearly), making the strategy to detect potential causes fairly weak. What the variables can say should be a bit more detailed to provide a stronger basement and state potential hypothesis to be tested. The selected variables do show changes in current water extent (not inundation) over the period but also channel behaviour (shifting vs non-shifting). Water extent provides an information on current flow conditions (not inundation) and can inform on changes in hydrological or morphological signal (narrowing = lower current flows or regulation/embankment/diversion ; widening = reservoir or higher current flows - interbasin transfer for example) AND channel morphodynamics (not change)... linked to channel shifting within the floodplains corresponding to specific multiple-thread channels (braided or anabranching/anastomosing) or free-meandering/wandering channels mainly (please note the morphodynamics can change as well... rivers can be less and less shifting or more and more). This is not clear in the title nor in the abstract. The key-issue is how to separate channel pattern types (stable and unstable) from real hydrological and morphological changes. If this is not done correctly, the analysis of potential drivers of change is weak. This is particularly true with the case of stable channels related to developed countries. Most of temperate and/or lowland systems over 40 years are stable independently of any infrastructures. The Amazone or the Congo are good examples. The widening related to new reservoirs is convincing because it is fairly easy to locate new dams and their reservoirs. The problem is much trickier with the potential climate change effect and the paragraph L203 -220 is not convincing at all. Again, there is a need to remove what is linked to the river behavior (there are many braided shifting rivers in Tibet and also new reservoirs) to explore this issue (what about west Africa or south Europe, why north Canada is not concerned?).

Abstract should be adjusted to better understand what is observed, what are the potential interpretations, what are the most important changes or channel behaviors.

Terms need to be fixed. Inundation or braided index are not the appropriate terms and this can provide confusions and misunderstanding in terms of drivers (see some of my comments on the drafts).

The number of available images (observations) between the first period and the second is significantly different (roughly it doubled over the period). It means the probability to observe more extreme conditions (high or low flow conditions) is not constant and this may affect the comparison of the two selected time windows. This could be discussed and better assessed according to the hydrological regimes/regions.

A list of potential drivers is established, but it is still very difficult to hierarch them. One of the reasons is linked to the fact that PI/PD and PGS are analysed separately so that it is difficult to separate what is linked to the geomorphic functioning (e.g., channel shifting) and what is linked to real changes (narrowing vs widening due to hydrology or morphological modifications).

I provided a lot of comments on the two manuscripts that should help the authors to understand the main key points I listed above.

Reviewer #2 (Remarks to the Author):

The manuscript presents an interesting analysis of river inundation extent changes from a global perspective using satellite images and correlating tendencies with climate change (precipitation changes, meltwater and global aridity), anthropic factors (relating to nightlight to identify developed countries, river embankments or dam construction and other human interventions) and geomorphological factors.

I consider that this manuscript has a very high value and is very appropriate to be published nonetheless I have some suggestions or comments.

The analysis compares the mean river inundation situation of the period 1984-1999 (15 years) to that of 2000-2018 (18 years). Therefore it can not be stated that the analysis is about "global river changes of the past nearly 40 years". It is just of the last 18 years, as the starting value is that of the mean extent of whole sub-period of 1984-1999, so it is compared with that of the last 18 years, so that is the period for which we are assessing if it has been a widening, narrowing or stabilization of inundation.

I do not see clearly explained how OCI value is obtained (-100 to 100).

Some specific comments:

Line 55. Maybe it could be find a more appropriate reference instead of "19".

Lines 114-115. The percentages do not correspond to the absolute data, I might be wrong but it is 14.7% instead of 18.9% and 5.5% instead of 7.1% according to each "n".

Line 365. "..., we chosen ..." substitute with "..., we chose..."

Figure 1b (extensive to the same graphs in other Figures). I suggest changing the title of the left vertical axis from "Count" to "Number of basins", as it is more clear.

Figure 3 h, i and j. It would be clearer if each graph is named with its region name or even include these letters (h, i, j) in the Figure 3f.

Figure S1. There is a mistake in the legend of a, c and e; "Perentage" it should be "Percentage" (also in Fig. S2, S3. It should be included the reference to a, c, e in the Figure captions. In my opinion this Figure should be in the main manuscript.

Figure S2. Why not include the name of the rivers in a, b and c figures?

Figure S12. I would keep the same scale for the graphs (b and c). b ranges from 0 to 0.6 and c from 0 to 0.2. In order to be comparable it should be the same one.

This gives the idea of more increasing than decreasing.

Figure S16. Legend it is confusing because of the the negative symbol and the hyphen, try to separate more.

Table S1. If I am not wrong, the table caption has a mistake: "..., and significantly decreased (PSI) inundation frequency.." it is "increased".

Reviewer #3 (Remarks to the Author):

I read this paper with great interest - it is a very ambitious exercise in remote sensing, leveraging a number of tools and global datasets to deliver a summary picture of change in major rivers around the world. It seems to be a natural outgrowth of earlier work by Allen and Pavlevsky (2018), introducing a temporal dimension to the large-scale mapping of river width and inundation.

I found that while the paper has produced some very nice plots and summaries of global inundation trends, both the introduction and conclusion did not distinguish on what is new and novel in this work and focus reader attention on a specific question. Clearly the paper shows a strong methodological advance, but I found that this does not clearly emerge as the focus of the paper. Many of the patterns identified in the results are consistent with previous works, and indeed most of the insights are confirmed by appeal to existing studies and literature. These broad trends are largely understood, although this is a novel way of detecting and displaying these global patterns.

I found there were a few key problems with the presentation of the work:

(1) There are a multitude of processes responsible for changes in river width, but only the broadest of reasons are addressed. The result is a fairly broad and diffuse statement on human interference and climate change. There is not a focused conclusion that emerges from this.

(2) Error is not addressed in the main paper, but is largely relegated to (brief) treatment in the supplementary material and previous work. What are the chances of clear and continuous coverage of each river each year? How are the seasonal effects (including winter ice, nival conditions, and veg changes) controlled for? The ratio of width and depth for various rivers will have a strong influence on the nature of inundation, so some rivers are far more likely to show variability. The examples provided in Figure 3b-e are not subtle ones. Without at least a few examples of the robustness of the method - how often the technique gets it right (local verification), particularly in smaller systems. I should think it would be of interest to the readership to better understand the precision of the technique, and what thresholds can be found at the tipping points between 'stable' and 'decreasing', for instance. Given that error is not clearly demarcated or addressed in this remote sensing study, I do not know how much confidence to invest in the work that follows.

(3) the magnitude of change is relative, and the implications of these changes are difficult to convey. In some regions, variability is cyclical and not unexpected. In others, the change is profound and irreversible. The percentage of moderate versus 'significant' change is not given much context. From this perspective, the paper seems unable to offer new and nuanced insights into the nature of decadal river change. Despite this problem, the authors have done a good job of reviewing existing literatures to confirm and contextualize the findings, showing that results are consistent with current understandings of global river change.

My overall sense is that the major contribution in this work is the methodology behind it, and not the patterns of channel change summary reported in the core paper. As someone with an interest in patterns of river change, I found much more interesting material in the appendix and supplementary sections than the broad and fuzzy generalizations in the main work. The tremendous effort that has gone into the geospatial change detection is much better reflected in the tools and techniques, rather than the description of changes, which are broadly known, but have not previously been summarized with this kind of detailed, spatially-explicit method.

I believe that the work would be better tailored for a premiere remote sensing journal, where the readership will better appreciate the innovation that is represented here. I would encourage the authors to merge the core and supplementary material for a more expansive treatment of the technique, better delineation of a 'research question', explanation of some of the errors/pitfalls encountered, and deeper perspective on the implications of change in the various large river systems. In its abbreviated form, I think you are doing something of a disservice to your efforts in algorithm development and application.

REVIEWER COMMENTS

Reviewer #1 (Remarks to the Author):

The MS entitled "Satellite-observed global river channel changes driven by anthropogenic and climate forcings" submitted for publication in Nature Communications explores changes in current water extent along fluvial network worldwide between mid-1980s to 2018 and factors controlling them. If the descriptive approach is convincing and really interesting, the causal approach, still exploratory to assess the "main" drivers, is difficult to follow and not convincing at all, except the part on dams. Indicated drivers are mainly interpretations and not facts and it is very difficult to really assess what is true and not true, and to hierarch these potential drivers.

Response: Thanks for the in-depth comments providing clear merits and demerits evaluation. We carefully considered the problems pointed out by the reviewer and performed further studies to improve the analysis, particularly regarding the hierarchy of different types of change and the potential drivers.

One of the main problem is linked to the fact that the explanatory variables are not clear (or introduced clearly), making the strategy to detect potential causes fairly weak. What the variables can say should be a bit more detailed to provide a stronger basement and state potential hypothesis to be tested. The selected variables do show changes in current water extent (not inundation) over the period but also channel behaviour (shifting vs non-shifting). Water extent provides an information on current flow conditions (not inundation) and can inform on changes in hydrological or morphological signal (narrowing = lower current flows or regulation/embankment/diversion ; widening = reservoir or higher current flows - interbasin transfer for example) AND channel morphodynamics (not change)... linked to channel shifting within the floodplains corresponding to specific multiple-thread channels (braided or anabranching/anastomosing) or free-meandering/wandering channels mainly (please note the morphodynamics can change as well... rivers can be less and less shifting or more and more). This is not clear in the title nor in the abstract. The key-issue is how to separate channel pattern types (stable and unstable) from real hydrological and morphological changes. If this is not done correctly, the analysis of potential drivers of change is weak. This is particularly true with the case of stable channels related to developed countries. Most of temperate and/or lowland systems over 40 years are stable independently of any infrastructures. The Amazone or the Congo are good examples. The widening related to new reservoirs is convincing because it is fairly easy to locate new dams and their reservoirs. The problem is much trickier with the potential climate change effect and the paragraph L203 -220 is not convincing at all. Again, there is a need to remove what is linked to the

river behavior (there are many braided shifting rivers in Tibet and also new reservoirs) to explore this issue (what about west Africa or south Europe, why north Canada is not concerned?).

Response: We sincerely appreciate your incisive discussions about the indication of variables and the associated scenarios of water extent changes. The insightful comments and constructive suggestions inspired us to solve the key issue - how to separate different types of signals in the detected water extent changes. This is a challenging issue on a global scale, and we are lacking literature reference to do so. However, we did find that river pattern shifting and morphological changes were evident on the OCI map. We developed an expert-based protocol streamlining the manual interpretation of morphological dynamics and employed a machine learning approach to identify morphological changes globally (Fig. S2, S3, S4; Fig. 1). As stated in the summary above, we separated different types of signals, morphological changes, dam-related river expansion and hydrological signals. Regarding morphological changes, we understand that they can be natural evolutions or can change as well (more or less shifting); however, we cannot make further identifications or conclusions based on the time span of current observations. In other words, our results represent an observation of global river morphological changes from 2000-2018 compared to 1984-1999. The pattern of morphological changes and potential driving factors were discussed in the first section of the results. The concerning points are addressed as follows.

Explanatory variables: According to the suggestion, we briefly described the explanatory variables and the indication in the introduction and in more details in the first section of the Methods. Specifically, we introduced how the inundation frequency changes were calculated and the meaning of the different classes. We stated potential hypothesis on the pattern of frequency changes regarding morphological dynamics, dams, and hydrological signals.

Stable channels: We agree that the stability of river water extent is determined by many factors and infrastructures are one of the contributors. Our results of river stability (Fig. 8) support that ‘Most of temperate and/or lowland systems over 40 years are stable independently of any infrastructures. The Amazon or the Congo are good examples.’ To demonstrate the effect of infrastructures, we revised our analysis by excluding basins where population density was very low (remote areas where human interferences are not significant). This means that river embankment infrastructures played a role in stabilizing river flow regimes in the major residence of the human being (excluding remote areas).

Climate change effect (paragraph L203 -220): The updated analysis focuses on the hydrological signal after removing morphological changes and dam-related water expansion. Obvious widening in the Tibetan Plateau and eastern Siberian can be observed on the map (Fig.3, Fig.5). We performed hotspot analysis to present the most prominent hydrological signal globally and

provided a detailed analysis of changes of climate variables in these regions (Fig. 6).

Abstract should be adjusted to better understand what is observed, what are the potential interpretations, what are the most important changes or channel behaviors.

Response: Thanks for the suggestion. We revised the abstract to better reflect the supported findings (facts) and the implications.

Terms need to be fixed. Inundation or braided index are not the appropriate terms and this can provide confusions and misunderstanding in terms of drivers (see some of my comments on the drafts).

Response: We agree with this suggestion. We avoided the usage of ‘Inundation extent’ or ‘braided index’ and replaced with “river water extent”, “river flow regimes”, “number of channels” instead. Thanks for the detailed comments. We made corrections accordingly.

The number of available images (observations) between the first period and the second is significantly different (roughly it doubled over the period). It means the probability to observe more extreme conditions (high or low flow conditions) is not constant and this may affect the comparison of the two selected time windows. This could be discussed and better assessed according to the hydrological regimes/regions.

Response: This is truly the known limitation of Landsat-based observations. We acknowledge that the uneven temporal coverage of the Landsat imagery introduced uncertainties and ambiguity in the OCI-based changes. Regarding the uncertainties and validation, we performed experiments from three aspects and presented them in the “Uncertainties” section. First, the robustness of frequency change patterns is illustrated from cases of different hydrological regimes/regions (Fig. S19). Second, we compared the the number of Landsat observations in each moth of the year during the first and the latter epoch and concluded no obvious seasonal bias (Fig. S20). Third, discharge measurements at gauging stations were used to evaluate the reliability of our statistics (Fig. S21). Although extreme conditions can affect the frequency statistics, we suppose that statistics essentially represent comparisons of the mean situations in the two periods.

A list of potential drivers is established, but it is still very difficult to hierarch them. One of the reasons is linked to the fact that PI/PD and PGS are analysed separately so that it is difficult to separate what is liked to the geomorphic functioning (e.g., channel shifting) and what is linked to real changes (narrowing vs widening due to hydrology or morphological modifications).

Response: We agree that separate analysis of PGS/PI/PD leads to fuzzy impressions without a clear picture of the overall change patterns. Therefore, in addition to separating different types of river

changes, we modified our analysis and presentation logic. Regarding the hydrological signal, we first presented the general patterns (which signal (increase, decrease, stability) is dominant in different regions/basins, Fig. 3, Fig. 4), and then focused on the most prominent changes (increase/decrease, Fig.5) and lastly the pattern of river stability (Fig. 8).

I provided a lot of comments on the two manuscripts that should help the authors to understand the main key points I listed above.

Response: Sincere thanks for the detailed comments. We reviewed them carefully and made corrections accordingly.

Some specific comments in the Manuscript:

Line25-26: "almost 40% of the global river area experienced obvious changes..." ---it would be good to indicate the total river length studied because it is 40% of a river network that is far from being the whole one due to the rough resolution of landsat images.

Response: Thanks for pointing out the confusing details. We primarily used river area rather than river length in the statistics. To be clear, we explicitly explained the metric of 'river area' in the first section of the Methods part, and gave the exact statistics (2,097,799 km by length and 769,390 km² by area) in the abstract and the main text. Our analysis is primarily built on the Landsat-based global river network (SWORD) and observations. We agree that the resolution of Landsat images limits the ability of a more complete river network mapping (e.g, river width < 30 m, Allen & Pavelsky, 2018).

Line27: "and evolving water policies" ---not clear / explicit

Response: This statement is removed here and explained later. To be clear, we specified 'water abstraction' and 'water use management to replace the 'evolving water policies'.

Line29: "particularly the Tibetan Plateau and eastern Siberia" --- reservoirs are also important in this part of the world

Response: We revised our analysis for a separate analysis of different types of signals. With a focus on the hydrological signal, we excluded new reservoir-type river reaches and morphological dynamics from the analysis. The hotspots of river extent changes still include the Tibetan Plateau and eastern Siberia, which means that river widening is a strong signal for the two regions. The sentences are modified to clarify the new clues of analysis.

Line33: "...confirming the expected services of river infrastructure including levees and embankments" --

-(reviewer only highlighted)

Response: Regarding this concern, we rephrased the argument to reduce the confidence.

' River widths maintain relatively high stability in developed regions, which were probably associated with the effect of river infrastructure including levees and embankments on confining river flow regimes.

Line35: "...being a response to natural channel morphology,..." --- again, unclear. Why would channel morphodynamics change recently and in which terms?

Response: Our analysis of morphological dynamics has been revised according to the suggestion. Our work highlights a global map of regions where river morphological dynamics has been prominent on the basis of long-term remote sensing observations. In terms of the causes of morphological dynamics, we discussed potential drivers including natural factors (hydrological and geological setting), climate change and human impact. A global scale attribution of the morphological dynamics is challenging and more observations and modelling are required.

Line36-37: this is a view point, not supported by facts

Response: To strengthen this viewpoint, we provided a detailed analysis of the hydrological signal and its relationship with changes in climate variables (Fig. 6) and human interventions (exemplified in Fig. 7). These analyses are consistent with previous studies focusing on local regions (e.g., drying in northern India, central Eurasia, eastern Brazil, the middle and southern North America). Although we do not provide direct evidence for each case, current observations and studies support this argument.

Line70: "...inundation changes..." --- not inundation. "flow channel"

Response: The term has been modified according to the suggestion.

Line71: "identify" --- explore "potential drivers"

Response: The sentence was rephrased accordingly.

Line72: "...of two major state-of-the-art surface water databases." --- not only because to make your interpretation as strong as possible to explain your cause-effect relationship you use other information. The paper describes and provides potential explanations

Response: We revised this sentence to highlight the method of deriving global river extent changes, and the analysis (cause-effect relationship) were summarized at the end of the introduction.

Line81: "...dynamics of rivers..." ---unclear

Response: This sentence was revised to improve clarity. *"To retrieve river extents we implemented..."*

Line84: "...probability of changes..." ---what kind of changes do you observe: change of the channel position in the floodplain, change in water extent due to hydrological or morphological changes. Clear need to explain properly what you can potentially see with this indicator and identify potential hypothesis you test.

Response: This is a good suggestion. We elaborated the indication of the used metrics (frequency changes vs. water extent changes) and the associated potential hypothesis to be tested. A detailed description is also supplemented in the Methods section. *"For rivers, positive (negative) values on the OCI map indicate increase (decrease) of water flow frequency, informing changes in river extent which can be associated with hydrological signals (flow conditions: widening/higher or narrowing/lower) or morphological changes in river platforms."*

Line88-89: need to be clarified

Response: Detailed description of how OCI changes are related to the water extent changes and river morphological dynamics is provided in the Methods section (Statistical variables and the indication).

Line89-90: "pattern of river channel evolution" ---unclear again

Response: Here, we mean the type and pattern of river extent changes (widening or narrowing, or morphological dynamics). The statement was revised for clarity.

Line96: "river channel changes" ---changes in flow extent along the channel network

Response: The suggested term is to the point. Throughout the manuscript, we updated the term 'river channel changes' to 'river extent changes' or 'river flow regime changes'.

Line97: "...inundation..." ---unclear again. Do you explore "mean flow" conditions with your indicator or "extreme flows - minimum and maximum" including inundation. This must be said here to clarify what is observed.

Response: According to the statistical method of the OCI database, the indicator (OCI value, namely the frequency changes) refers to the comparisons of mean flow conditions in normal cases (observations were not biased toward certain seasons in each period). The statement was revised accordingly.

Line97: "...638,887 km² of river area..." ---river length would be more meaningful. What is the frequency of the water extent changes in this total area?!

Response: We provided statistics of different classes of frequency changes for the global river area. "Globally, the percentage of the significant increase (moderate increase) reaches 9.0% (8.6%), higher than that of significance decrease (4.8%) (moderate decrease: 7.4%). Meanwhile, 70.2% of the river areas are relatively stable in water extent."

Line101: "river" ---delete

Response: The statement was rephrased for clarity.

Line101: "or flow" ---delete

Response: The statement was rephrased for clarity.

Line108: "...river channel changes..." ---patterns of flooding? What it is observed here is not really a "channel change"

Response: The term 'river channel changes' was misused and we corrected the usage accordingly.

Line127: "river channel" ---delete

Response: The term 'river channel changes' was corrected to 'river extent changes.'

Line127: "...changes..." ---flow area ... changes

Response: The term 'river channel changes' was corrected to 'river extent changes.'

Line129: "...increased inundation frequency than decreased" ---it is very confusing because you mix different things. Net decrease, net increase and a balance in the case of channel shifting... and considering separately increase and decrease only does not really help to understand what's going on mixing effective hydrological and morphological changes and channel behaviors.

Response: We agree that the mixture of morphological dynamics in the analysis will disturb the understanding of the hydrological signal. In the revision, we devoted a lot of time and effort to separate the morphological signal, which has never been done on a global scale. By identifying and separating different types of river extent changes, we enabled a clear focus on the hydrological signal here. Given that widening and narrowing signals probably coexist in a single basin due to high variability of river changes, we also considered the net increase/decrease metrics in our revised

analysis to evaluate the magnitude of the primary signal. See the section titled 'Patterns in megabasins' for a detailed revision of all changes.

Line130-131: "...top three in percentage of increase (63.0%, 43.0%, and 46.5%, respectively)..." --- Typically they do not widen, they just move in their floodplain

Response: After revision (with morphological signal separated), the analysis here is mainly about the hydrological signal.

Line154: "...changes..." ---change in water area extent and river behavior

Response: This statement was removed in the revised manuscript. We took this advice and revised related expression in other parts of the manuscript.

Line154: "...first-order attributions..." ---how do you consider they are first-order? Can you provide the river length concerned by each of them? You may state explicit hypothesis and show how you can validate them.

Response: We have revised the logic of the results according to the suggestions. After revision, we retrieved hotspots of river changes (excluding dam-related changes and morphological dynamics) and provided analysis of the causes.

Line155: "...significant increases in inundation frequency..." ---water extent

Response: Thanks for the detailed advice. We carefully reviewed the phrase 'inundation frequency' and made revisions accordingly.

Line165: "...11,404 km² of the intensified inundation area." ---a table summerizing the contribution of the different potential drivers and the unexplained part would help to better hierarch them.

Response: Thanks for the suggestion. Accordingly, we recognize dam-related river expansion as a particular form of river extent changes and analyzed it separately. In the revised analysis of this issue, we provided information about the total area of new reservoir-type river reaches, the percentage of expanded flow area, and its contribution to river widening signal on different spatial scales.

Line166: "...46.4% of the total river area increase..." ---it is really high?!

Response: We are sorry for the confusing point here. Here 46.4% is the ratio of dam-related expanded river extent to the total expanded river extent globally. The expanded river extent refers

to the part of river areas marked as significantly increased frequency (OCI > 75%). The high ratio (46.4%) indeed reflects the remarkable contribution of new dams to the global river widening signal. In our revised analysis, we refined our analysis of reservoirs (new-reservoir inventory intersecting SWORD river network) and excluded basins dominated by morphological dynamics in the statics of global widening signal. The updated ratio (31.9%) still support new dams as a major contributor to global river widening.

Line184: "Water policy promoted inundation recovery..." ---not clear?

Response: Here we originally mean that improved water use management helped the recovery of normal water flow in the region. This section was revised and merged in our analysis of the human impact on river extent changes.

Line189: "...no-flow cutoff..." ---unclear

Response: We provided detailed definition of no-flow cutoff in the main text here. *'The high percentage of increased flow coverage in this region is essentially a recovery from no-flow cutoff (Less than 1 m³/s runoff at the Lijin hydrological station situated at the lowest reach of the Yellow River).'*

Line195: "...improved during the early 21st century" ---why is it so? Explanation fairly unclear

Response: We provided more detailed explanations by summarizing the main levers the local authorities used to improve water use management in the Yellow River basin. Related studies are also cited for further references of this topic.

Line210-211: "...the percentages of increased inundation extents are 67.5%, 39.2%, and 51.0%, respectively" ---you need to remove what is linked to the river behavior (many braided shifting rivers in Tibet and also new reservoirs) to explore this issue.

Response: Thanks for the notes. In the updated manuscript, we provided separated analysis of morphological behavior and hydrological signal.

Line214: "...precipitation increasing at 11.62, 40.32, and 25.19 mm/year" ---This is a total annual increase but it has nothing to do with flooding that is related to maximum rainfall events or ice-melt

Response: We agree that extreme precipitation events can cause flooding events and consequently impact on the water frequency changes. However, the frequency change on the OCI map is the mean difference of the overall frequency between the two epochs. Therefore, individual events have limited influences on the long-term statistics. In other words, we inspected that the OCI map reflects

comparisons of mean situations in two periods, and comparison of the overall changes of climate variables is appropriate.

Line229: "...outlined three regions (h-j)." ---indicate each of them explicitly

Response: This figure was revised and the specifications were supplemented.

Line231: "...arid regions" ---The map does not only show narrowing in arid regions? Do you see other potential drivers? ... again potentially water diversion in side canals for irrigation or river straightening/embankment can be also potential drivers. again need to consider narrowing vs widening to better separate real hydrological/morphological change from river behavior.

Response: We agree that morphological change should be separated from hydrological change. The analysis was updated after such a classification. It is true that decrease signal exhibited high spatial variability and different levels of narrowing occurred in different parts of the world. Our point here is that most of the prominent decrease signal concentrated in the arid regions in the interior continental interiors (e.g., central North America, central South America, middle Eura-Asia). And the causes are probably associated with both climatic factors and human interventions including water diversion and abstraction through canals.

Line253: "...groundwater abstraction..." --- groundwater? Can be completely independent of river network. We can expect water diversion (also related to damming) can play a significant role.

Response: We agree that groundwater and river network are different things, but we politely disagree that they are independent. In many regions, groundwater interchanges with water in rivers and lakes through base flow, particularly in arid regions (simple illustration at <https://books.gw-project.org/groundwater-in-our-water-cycle/chapter/groundwater-connection-with-streams/>).

Excessive abstraction of groundwater can lead to declined groundwater level, inverted stream-aquifer gradient and stream leakage (Krueger et al., 2017). In the revised analysis, stream narrowing in arid regions is still remarkable, despite the exclusion of new reservoir-type river reaches. We speculate that groundwater abstraction played an important role in influencing river water levels on the basis of facts reported in previous studies.

Line271: "...may be correlated with the socio-economic development level..." --- big big assumption... without understanding/Indicating river patterns (stable vs laterally shifting) are first linked to bioclimatic regions. In temperate and/or lowland areas, river patterns are fairly stable.

Response: It is agreed that hydroclimate features are the first-order control of free-flowing rivers. However, there are emerging evidence that the natural river systems have been disturbed/fragmented by anthropogenic activities (Best 2019; Grill et al., 2019; Belletti et al., 2020). We raised the assumption here on the basis of the pattern of river extent stability (the percent of stable class in each basin) and evaluated it with indicators of socio-economic development (nighttime lights density and population density).

Line273: not clear here what is the cause and what is the consequence. Better for humans to establish along stable channels.

Response: We intended to compare the pattern of stability with the pattern of nighttime light intensity, with no implication of the cause and the consequence. To avoid confusion, we rephrased the sentence and provided further explanations that human projects (as indicated by the nighttime light intensity) lead to stable river extents.

Line279: "...levees to constrain lateral floods" --- we do not focus here on flooding or inundation but on current flow area. This can't be a driver at all.

Response: We are sorry for the misleading information. Here we mean that the levees were built for the purpose of constraining lateral floods. We rephrased the sentence to avoid such confusion.

Line287: "...bank erosions, thus substantially stabilizing river..." --- dikes and bank protections are (or can be) two distinct infrastructures. Very unclear this point. We may hypothesis that in developed countries, we should expect to have fairly more shifting rivers in the piedmont areas due to bank protections to prevent bank erosion... but you need first to locate where are potential shifting rivers and which part of them do not shift anymore. At this stage, the demonstration is far from being convincing.

Response: We agree with your opinion that the function of man-built river infrastructures should be evaluated with more rigorous reasoning and evidence. However, a rigorous justification here can be impractical due to limited observations and understanding of which and how rivers evolve in response to natural and human forcings. As suggested, historical observations (several decades or hundreds long) are required to show the behavior of rivers before and after setting of infrastructures, but availability of such observations on a global scale is the key issue unsolved. Here we showed the pattern of river stability related to the development level of human societies from a perspective of region scales, and it may be impossible to go to the details for individual rivers or reaches as observations (e.g., detailed river infrastructures) are incomplete. The demonstration here offers a simple and straightforward explanation for the function of river embankment infrastructures, as

they normally consolidate the river banks, mitigating the erosion and sedimentation processes and therefore leading to stable river flow regimes. We rephrased the sentences to clarify the simplified assumption here.

Line297: "...can be characterized by four major metrics..." --- different things are mixed here. You may clarify if you want to analyse river pattern (planform types - multiple vs single thread, sinuosity ...) and the potential drivers explaining these patterns (e.g. slope)

Response: The analysis was updated according to the suggestion. In characterizing river pattern, we employed two metrics in the classification: the number of channels and sinuosity.

Line298: "mean flow" ---(reviewer only highlighted)

Response: This means the average width of river reaches (river width when the discharge is around the seasonal means). We used this term according to the definition in the GRWL database (Allen & Pavelsky, 2018).

Line299-300: "...where rivers have high sinuosity and steep hydraulic gradients..." --- implicitly here, you are looking for a specific highly shifting channel patterns : the free meandering patterns. This may be said more explicitly. Another highly shifting pattern, the most important one, is the braided pattern usually located a bit upstream.

Response: We appreciate the excellent idea of identifying different styles of morphological changes. We devoted a framework to interpret such signals on the basis of the new river database and the OCI map showing river extent changes.

Line304-305: "stable channels are associated with high braiding index and wide rivers, which are the general characteristics of midstream or downstream river..." --- not braided... multiple-thread. Braided systems are located in mountain plains or piedmonts, always upstream of the shifting meandering systems

Response: Thanks for pointing out the details. The analysis was revised in the updated manuscript.

Line308: "...Brahmaputra, the Ganges, the Indus, and the Irrawaddy..." --- these are the most important braided systems on earth.

Response: We agree that they are important braided systems on earth. The results of our updated analysis support that the five rivers are the major compositions of global river systems showing morphological changes in the recent decades.

Line331: "...and water use management" --- not convincing

Response: We modified discussions on the impact of water use management on river extent changes to provide more details. Specifically, we discussed that prominent river narrowing in several arid regions of interior continents is connected with increasing water demands and excessive groundwater abstraction; This is in contrast to river flow recovery in the Yellow River basin due to improved water use management. Regarding how water use management improved water flow in the Yellow River, we provided more background information and literature. The Yellow River case demonstrated that adjusted water use management can recover normal river flow, which provides a salutary lesson for the high water-demand regions.

Line332: "climate forcing, including shifting precipitation patterns and cryospheric response..." --- again not convincing

Response: To support this argument, we provided a more detailed analysis of precipitation/evapotranspiration changes in the past four decades in the results section. We discussed the role of cryospheric response by relying on evidence and studies provided in previous literature.

Line339-340: not clear. Why would it be so?

Response: We provided global maps of river flow regimes changes, contributing to the understanding of river morphological dynamics, new dams on rivers and different levels of 'narrowing' and 'widening' signal. Our attribution analysis highlights the driver of climate and anthropogenic forcings to river widening/narrowing in the recent decades. The maps provide satellite-based evidence of river extent changes, which help guidance on where river restoration and improved water use managements are urgent, such as in the arid and continental interiors with strong narrowing signal.

Line351: "...braided rivers..." ---multi-thread channels not braided

Response: The term was revised accordingly.

Line381-382: "...homologous pairs of months between two epochs." ---(reviewer only highlighted)

Response: This statement means that frequency changes were compared between homologous months, and then averaged among pairs.

Line387: "...inundation extents" --- current flow extent may be more appropriate than inundation. Inundation may concern an area that is usually dry and can be for a short period of time inundated. What we see here is different. When the channel moves, it abandons some areas and occupies others. What we see then is the current flow channel that change its position. Nothing to do with inundation.

Response: Thanks for the detailed explanation and the suggestion. We reviewed the use of 'inundation extents' through the manuscript and revised where necessary to assure the appropriate usage. We agree that river extent or river flow regime is more appropriate than inundation extents.

Some specific comments in the Supplementary Text:

Line28: "...or migration downstream" --- usually when sediment supply is reduced, channel migration is also reduced, both are linked. This hypothesis downstream reservoirs is not tested

Response: Thanks for notifying this point. In this section, we talk about some straightforward scenarios but not facts. We understand concerns about the untested hypothesis. Thus, we revised the description in this section (moved to the methods part).

Line31-32: This hypothesis is not explicitly tested.

Response: We understand concerns about untested hypothesis. The related description was weakened.

Line35-36: "...for flood prevention, which generally leads to reinforced stability of the inundation area (widths)..." --- This can't be studied with the data. There is a big confusion with the term inundation.

Response: We rephrased the sentence and particularly the term 'inundation' here and elsewhere.

Line60-64: unclear why this is here.

Response: These statements were removed to appropriate places in the main text.

Figure S1 a, b, c: "percentage" ---percentage

Response: We are sorry for the misspelling. The word was checked and corrected. As this information has been directly or explicated presented in the main text, we removed this figure in the supplementary file.

Line87: "The case of Yellow River cutoff" ---what is cut-off? Unclear

Response: The cut-off means no river flow in parts of the river reaches. This figure was moved to the main text and the description was revised for clarity.

Line94: "...braiding index..." --- need to explain how this is calculated. We expect a significant increase/decrease shifting with BI increases and this is not the case. How would you interpret this?

Response: The braiding index is the number of channels along the cross section of the rivers (Allen & Pavelsky, 2018; Altenau et al. 2021). We calculated the average number of channels for each basin. Note that the braiding index only gives information on the number of channels (distinguish of single-channel from multiple-channel) but not indicating the type of the river platforms (braided, anabranching or anastomosing). To avoid confusion, we corrected the term 'braiding index' to 'number of channels'.

The non-shifting signal with larger number of channels may be explained by several factors. Firstly, the statistics here did not separate morphological dynamics from hydrological signals. This may cause mixed signals as you suggested. Secondly, this could mean that the number of channels alone is not a good indicator of the river change patterns, as many other features such as sinuosity, mean width and other non-quantified features may play an important role. Thirdly, river extent changes showed high spatial variability globally. Considering these factors, we removed the overall statistics and this figure in the updated manuscript.

Line94: "...Amazon River." --- it is not a braided river in the lowland area but an anabranching or anastomosing system (multi-thread non-shifting channel pattern).

Response: Thanks for the detailed information. We avoided definition on the types of specific river systems in the revised main text.

Line98: "...Congo river." ---I am surprised that in the Congo basin we have braided rivers... Perhaps you mean multiple-thread channels not braided channels. The geomorphic functioning can be very different between an anastomosing and a braided system. The first is very stable whereas the second is actively shifting. If you mixt the two, you may see nothing.

Response: Thanks for this interesting discussion. We agree that there are different classifications regarding the types of river platform. Here we took a simple classification of two-types of rivers, the multiple-thread versus single-tread, based on the metric of 'number of channels'. From this definition, braided channels belong to multiple-thread channels but not all multiple-thread channels are braided. We referred to relevant literature and revised the use of such terms.

Figure S9 a: "Buffer" --- here again, need to explain what is a "buffer distance".

Response: The buffer distance refers to the distance around the input river centerlines that will be buffered to represent the statistical river extent. In river database (SWORD or GRWL), only centerlines are provided. From river centerlines to river buffer zones, the key parameter is the buffer distance. We explained it a little bit in the Methods section.

Figure S9 d: "percentage" ---(reviewer only highlighted)

Response: We are sorry for the misspelling. The word was checked and corrected.

Line111: "...at 30-m distance..." --- unclear. It means longitudinal distance?

Response: Yes, it is longitudinal distance. The river width is measured at each 30-m distance (one pixel of the Landsat imagery) along the river centerlines. This attribute is contained in the GRWL database. The flow chart was revised as we updated the river network with SWORD.

Line115: "...production of river inundation frequency change map" --- It can be SI or SD. Both are considered separately. Here only SI is shown for illustration. Please clarify.

Response: Thanks for the detailed point. We revised the description for clarity.

Line115: "...river..." ---(reviewer only highlighted)

Response: Same to the previous point.

References:

- Allen, G. H. & Pavelsky, T. M. Global extent of rivers and streams. *Science* **361**, eaat0636 (2018).
- Altenau EH, et al. The Surface Water and Ocean Topography (SWOT) Mission River Database (SWORD): A global river network for satellite data products. *Water Resources Research* **57**, e2021WR030054 (2021).
- Belletti, B. *et al.* More than one million barriers fragment Europe's rivers. *Nature* **588**, 436–441 (2020).
- Best, J. Anthropogenic stresses on the world's big rivers. *Nat Geosci* **12**, 7–21 (2019).
- Grill, G. *et al.* Mapping the world's free-flowing rivers. *Nature* **569**, 215–221 (2019).
- Krueger ES, et al. Human factors were dominant drivers of record low streamflow to a surface water irrigation district in the US southern Great Plains. *Agricultural water management* **185**, 93-104 (2017).

Reviewer #2 (Remarks to the Author):

The manuscript presents an interesting analysis of river inundation extent changes from a global perspective using satellite images and correlating tendencies with climate change (precipitation changes, meltwater and global aridity), anthropic factors (relating to nightlight to identify developed countries, river embankments or dam construction and other human interventions) and geomorphological factors.

Response: Thank you so much for the summary and commendation. We carefully evaluated all the comments below and made a thorough revision to further improve the clarity and rigorism of the manuscript.

I consider that this manuscript has a very high value and is very appropriate to be published nonetheless I have some suggestions or comments.

Response: Thanks again. We have carefully addressed each of these suggestions and comments.

The analysis compares the mean river inundation situation of the period 1984-1999 (15 years) to that of 2000-2018 (18 years). Therefore it can not be stated that the analysis is about "global river changes of the past nearly 40 years". It is just of the last 18 years, as the starting value is that of the mean extent of whole sub-period of 1984-1999, so it is compared with that of the last 18 years, so that is the period for which we are assessing if it has been a widening, narrowing or stabilization of inundation.

Response: This is a good point, and we agree our analysis is about decadal changes in river extents. We revised the statements in the abstract, introduction, and conclusion sections (from 'the past nearly 40 years' to 'the early 21st century or 'during 2000-2018') to avoid confusion.

I do not see clearly explained how OCI value is obtained (-100 to 100).

Response: We are sorry for the missing details. We revised the Methods section about the method of deriving OCI value. 'The OCI map was created by averaging all surface water occurrence differences derived from homologous pairs of months between the two epochs (16 March 1984–31 December 1999 and 1 January 2000–31 December 2018) from the GSW database'.

Some specific comments:

Line 55. Maybe it could be find a more appropriate reference instead of "19".

Response: Thanks for the suggestion. We removed this reference and replaced it with an appropriate one (Best 2019) to support the influences of human impact on river systems.

Lines 114-115. The percentages do not correspond to the absolute data, I might be wrong but it is 14.7% instead of 18.9% and 5.5% instead of 7.1% according to each "n".

Response: We are sorry for the confusing point. The percentage is the statistics by river area not by basin count, but only the number of basins was given. We updated the statistics here (note that we updated the river network data and the analysis method), and explicitly specified if the percentage is by area or by count in the main text.

Line 365. "..., we chosen ..." substitute with "..., we chose..."

Response: Thanks for notifying the error. It was corrected.

Figure 1b (extensive to the same graphs in other Figures). I suggest changing the title of the left vertical axis from "Count" to "Number of basins", as it is more clear.

Response: This figure was removed as much of the information has been shown in the figures in the main text.

Figure 3 h, i and j. It would be clearer if each graph is named with its region name or even include these letters (h, i, j) in the Figure 3f.

Response: This figure was deleted in the revised manuscript. We show hotspots of increase and decrease signal in Fig. 5 in the main text. Each hotspot was represented with a letter for clear reference.

Figure S1. There is a mistake in the legend of a, c and e; "Perentage" it should be "Percentage" (also in Fig. S2, S3. It should be included the reference to a, c, e in the Figure captions. In my opinion this Figure should be in the main manuscript.

Response: We are sorry for the misspellings. The errors were corrected. According to your suggestion, we synthesized the key information in these figures to the main manuscript. The Fig.3 in the main manuscript is based on information in the Figure S1. Zoom-in of the river widening and narrowing for typical regions was also move to Fig. 5 in the main manuscript.

Figure S2. Why not include the name of the rivers in a, b and c figures?

Response: We provided such information in the figures when necessary (Fig. 5).

Figure S12. I would keep the same scale for the graphs (b and c). b ranges from 0 to 0.6 and c from 0 to 0.2. In order to be comparable it should be the same one. This gives the idea of more increasing than

decreasing.

Response: This is a good idea. We updated the Fig. 4 according to your suggestion. As this figure contains excessive information related to Fig. 4, we removed it in this round of revision.

Figure S16. Legend it is confusing because of the the negative symbol and the hyphen, try to separate more.

Response: This figure was revised (now Fig. S19) as we provided more examples for the comparison. We avoided such issues in the revised figure.

Table S1. If I am not wrong, the table caption has a mistake: "..., and significantly decreased (PSI) inundation frequency.." it is "increased".

Response: Sorry for the mistake. It should be 'significantly increased'.

Reviewer #3 (Remarks to the Author):

I read this paper with great interest - it is a very ambitious exercise in remote sensing, leveraging a number of tools and global datasets to deliver a summary picture of change in major rivers around the world. It seems to be a natural outgrowth of earlier work by Allen and Pavlevsky (2018), introducing a temporal dimension to the large-scale mapping of river width and inundation.

Response: Thanks for the encouraging comments. The in-depth understanding inspired us to improve and polish the work. We carefully reviewed each comment and suggestion, and revised the manuscript accordingly. Specifically, we paid particular attention on refining the analysis of river extent changes for a clear ‘picture of change in major rivers’, enriching information about the techniques in the methodology, assessing the uncertainties and improving the presentation (figures and synthesis of supplementary information, highlights of the novel points in terms of methods and findings). Please refer to the response below and the letter of summary of changes for details.

I found that while the paper has produced some very nice plots and summaries of global inundation trends, both the introduction and conclusion did not distinguish on what is new and novel in this work and focus reader attention on a specific question. Clearly the paper shows a strong methodological advance, but I found that this does not clearly emerge as the focus of the paper. Many of the patterns identified in the results are consistent with previous works, and indeed most of the insights are confirmed by appeal to existing studies and literature. These broad trends are largely understood, although this is a novel way of detecting and displaying these global patterns.

Response: We sincerely appreciate your constructive feedback on the presentation of this work. We highly agree that we should appropriately clarify what is new in this work regarding both methodologies and insights into the scientific question. Accordingly, we modified the abstract, introduction and conclusions sections as well as new supporting figures to highlight our contributions in terms of methodologies and findings. Specifically, we emphasized that our results are based on a new framework (Fig. S1, S2, S3) for detecting and interpreting decadal river extent changes. The key methodological points include how to do the statistics on appropriate spatial scales and interpret the types of changes related to dams, morphological dynamics, and hydrological signals. In terms of findings, we argued that a global map of river morphological dynamics and a global quantification of new reservoir-type river reaches are novel contributions to the literature. Regarding the river widening and narrowing patterns, most of them have been confirmed or mentioned in previous works, but we provide the first satellite-based evidence of regional river widening in the Tibetan Plateau and eastern Siberia, which has only been indicated or speculated in previous studies. We

agree that this work emerges a novel way of detecting and displaying global patterns of river flow regime changes. We believe it contributes to an improved understanding of river changes in the scenario of climate change and intensified human interferences and spatially-explicit guidance for better prioritizing future river protection for sustainable development.

I found there were a few key problems with the presentation of the work:

(1) There are a multitude of processes responsible for changes in river width, but only the broadest of reasons are addressed. The result is a fairly broad and diffuse statement on human interference and climate change. There is not a focused conclusion that emerges from this.

Response: We agree that river extent changes are results of complicated processes driven by different forcings, including those of climate and anthropogenic, internal or external. To better hierarch multiple potential drivers, we devoted further efforts to classify different types of river changes, namely morphological dynamics, dam-related widening, and hydrological signals. Our analysis was therefore considerably refined in the revised manuscript, with a separate analysis of these different types of changes and the first-order attributions. We notably strengthened the presentation of the pattern of hydrological signals with hotspot analysis and clear evidence of climate change by referring to multiple climate datasets. We acknowledge the limitation that only the broadest reasons for these changes were addressed, as our focus was on global and broad scaled where river width changes exhibited high spatial variability. Besides, there is limited understanding regarding how river morphology evolves naturally or in response to external forcings. To answer the question, combined in-site and laboratory observations as well as numerical modeling approaches, are required, which is beyond the scope of this study.

(2) Error is not addressed in the main paper, but is largely relegated to (brief) treatment in the supplementary material and previous work. What are the chances of clear and continuous coverage of each river each year? How are the seasonal effects (including winter ice, nival conditions, and veg changes) controlled for? The ratio of width and depth for various rivers will have a strong influence on the nature of inundation, so some rivers are far more likely to show variability. The examples provided in Figure 3b-e are not subtle ones. Without at least a few examples of the robustness of the method - how often the technique gets it right (local verification), particularly in smaller systems. I should think it would be of interest to the readership to better understand the precision of the technique, and what thresholds can be found at the tipping points between 'stable' and 'decreasing', for instance. Given that error is not clearly demarcated or addressed in this remote sensing study, I do not know how much confidence to invest in the work that follows.

Response: We appreciate that the reviewer has considered many aspects of the uncertainties, which helped us deal with the tough issue. We, therefore, conducted further work to address these questions in the last section of the Methods part (Uncertainties). We addressed the uncertainties in three aspects: the robustness of the OCI statistics, the temporal coverage and the possible seasonal effect of Landsat observations, and how the statistics are related to in-situ river observations.

First, the robustness of frequency change patterns is illustrated from cases of different hydrological regimes/regions (Fig. S19). This confirms our expectations as the algorithm for water detection has been highly developed, and the OCI statistics took account of seasonality (Pekel et al. 2016). The major uncertainty is rooted in the common feature of Landsat-based observations, the uneven temporal coverage of the Landsat imagery, which means the number of Landsat imagery was lower in the 1980s (due to commercial management of the programme at that time) than that after 1990. With this feature, we suspect that the statistics during the first epoch might be more reflective of some regions' situations in the 1990s. Although the question about temporal coverage has been elaborated in the work of Pekel et al. (2016), we conducted further analysis to see if there was seasonal bias in the observations between the two epochs. Then as a second aspect, we compared the the number of Landsat observations in each month of the year during the first and the latter epoch and concluded no obvious seasonal bias (Fig. S20). Note that the number of observations in different months can differ, but this difference was generally consistent for a specific region over the years. Seasonal effects were also controlled as the frequency statistics (OCI) were compared in homologous pairs of months. Third, discharge measurements at gauging stations were used to evaluate the reliability of our statistics (Fig. S21). The results (Fig. S21) show an overall good correlation between decadal net changes in river extent and relative decadal annual discharge changes, given the differences in spatial scales (point-based versus region-based).

About river stability: We agree that 'The ratio of width and depth for various rivers will have a strong influence on the nature of inundation, so some rivers are far more likely to show variability'. This is also validated from our results that about half of the studied river reaches showed relative stability in flow extents, regardless of the types of river changes. Our work was essentially to detect where prominent changes occurred, the types of the changes, and the potential main drivers. Because currently, we know little information on the geometrics of rivers (such as the ratio of width and depth) globally, we cannot offer an in-depth analysis of the stability. We only provide a perspective on the stability from the general large-scale patterns but not the details of specific rivers or reaches. Additionally, we clarify that 'Figure 3b-e' in the original manuscript is about the examples of reservoirs widening river reaches and no indication of the stability (we missed something here?).

About thresholds of stability and changes: The thresholds of stability and changes should be considered from different spatial scales. For individual pixels along rivers, we consider that the classification of the OCI values (five classes: Significant increase, Moderate increase, Generally Stable, Moderate decrease, and Significant decrease) is quite reliable and robust (Pekel et al. 2016; Feng et al., 2019). On a broad spatial scale (e.g., level-6 basins), it is ambiguous whether it is stable, as different classes of changes generally coexist, which could reflect variabilities, morphological changes, and observation uncertainties. Essentially, we can view the changes (with morphological changes and new dams excluded) composited from different classes (increase, stable, decrease), as shown in Fig. 3, and identify the dominant signal. We proposed using the relative magnitude of net increase, which is the difference between areas of increase and decrease divided by the total river area ((increased area – decreased area)/total river area) to measure the dominance of different signals. This metric acts as a normalized variable for comparing changes in different regions.

(3) the magnitude of change is relative, and the implications of these changes are difficult to convey. In some regions, variability is cyclical and not unexpected. In others, the change is profound and irreversible. The percentage of moderate versus 'significant' change is not given much context. From this perspective, the paper seems unable to offer new and nuanced insights into the nature of decadal river change. Despite this problem, the authors have done a good job of reviewing existing literatures to confirm and contextualize the findings, showing that results are consistent with current understandings of global river change.

Response: Thanks for the insightful comments on the complexity of river changes. We provided specifications on the context of the used metrics and hypothesis in the introduction and methods part. Particularly, we review the 'significant' change as very high confidence of changes in water extents and flow conditions compared to 'moderate' changes, considering the variation and observation limitations. We set up a hypothesis that river flow regime changes, either of morphological dynamics or hydrological signals, can be detected by analyzing the pattern of frequency changes. Furthermore, we systemically considered different types of river extent changes and separated them using machine learning. The results present a global map of river morphological changes, a global assessment of reservoirs on river flow regime changes for the first time and a consistent map of river widening or narrowing signals that occurred in the early decades of the 21st century. We expect such information helps improve understanding of the pattern of decadal river water changes and contributes to better guidance for river protection. We acknowledge that future studies are needed to understand how rivers respond to climate and human forcings.

My overall sense is that the major contribution in this work is the methodology behind it, and not the patterns of channel change summary reported in the core paper. As someone with an interest in patterns of river change, I found much more interesting material in the appendix and supplementary sections than the broad and fuzzy generalizations in the main work. The tremendous effort that has gone into the geospatial change detection is much better reflected in the tools and techniques, rather than the description of changes, which are broadly known, but have not previously been summarized with this kind of detailed, spatially-explicit method.

Response: Thanks for pointing out the issue with detailed explanations. We seriously considered this problem and revised the logic and the core contents to avoid a total bias toward technical issues. Specifically, considerable changes have been made in describing the patterns to avoid broad and fuzzy generalizations. We supplemented detailed information and various supporting statistics to be as specific and clear as possible. As you suggested that there is interesting material in the appendix and supplementary sections, we synthesized information that could be interesting and helpful for understanding river changes into the main figures. However, as you suggest that the methodology is novel and should be highlighted, we have to achieve a balance between explaining the methods, hypothesis, and uncertainties and describing the patterns and findings. We enriched methodological information in the Methods part and put supporting figures and tables in the supplementary file.

I believe that the work would be better tailored for a premiere remote sensing journal, where the readership will better appreciate the innovation that is represented here. I would encourage the authors to merge the core and supplementary material for a more expansive treatment of the technique, better delineation of a 'research question', explanation of some of the errors/pitfalls encountered, and deeper perspective on the implications of change in the various large river systems. In its abbreviated form, I think you are doing something of a disservice to your efforts in algorithm development and application.

Response: We appreciate that you advocate the methodology highly. But generally, we believe this work appeals to a broad audience interested in river dynamics in response to climate change and direct human impacts. Although the methodology offers a new way of examining global river changes, the results clearly demonstrate the merits of the methodology by presenting an overall picture of different types of river extent change signals and the global pattern in recent decades. We understand that the previous manuscript presentation was weak in offering a clear and in-depth analysis of the results, so the findings seemed ambiguous and general. Therefore, we made substantial changes in the logic and contents, with efforts to identify different types of river extent changes and the hierarchy of multiple drivers more systematically. According to the suggestions, we also reorganized the materials in the supplementary files and methods for a clear and expansive

description of the methodology that would help future research on studying decadal river changes. Please refer to the summary of changes and the revised manuscript for detailed revisions.

References:

Pekel, J.-F., Cottam, A., Gorelick, N. & Belward, A. S. High-resolution mapping of global surface water and its long-term changes. *Nature* **540**, 418–422 (2016).

Feng, S. *et al.* Inland water bodies in China: Features discovered in the long-term satellite data. *Proc National Acad Sci* **116**, 25491–25496 (2019).

Dear reviewers,

Thank you very much for your help in reviewing and improving this manuscript. With full consideration of your concerning points and constructive comments, we performed a thorough revision to improve the analysis and presentation of this study. This takes longer than expected, and thanks for the extended time which allows for the completion of a better-quality revision. Here we explain and summarize the major changes below.

Logics of the main changes: One key issue raised by the reviewers is that our previous analysis did not hierarch multiple potential drivers of river extent change, which leads to a too general or fuzzy description of the river change patterns. In the original manuscript, we discussed river changes of increase, decrease, and stability separately and organized them into five different stories. In response to the reviewer's suggestions, we now hierarch multiple potential drivers by **separating different types of river flow regime changes: morphological dynamics, river expansion due to new dams, and hydrological signals (impacted by climate change and/or other ways of human activities)**. The challenge of this question lies in identifying the morphological dynamics that result in river extent changes, which require detailed information on the river morphological attributes. Therefore, we reprocessed all river statistical extent and improved the techniques of generating river extents with the informative attributes (e.g., meandering length, sinuosity, number of channels) according to the SWORD river database (the river network mostly inherits from the GRWL database). Finally, the schematic chart of the revised data processing and analysis is presented in Fig. S2 in the supplementary file.

Our analysis assumes that 1) morphological dynamics represent a particular style of river extent changes that are predominated by channel shifting rather than hydrological signals such as water extent widening or narrowing, 2) dam-related river expansion reflects notable river widening due to direct human intervention, 3) the remaining information reflects the general hydrological signal (widening/narrowing). Therefore, the logic of the results is changed from five stories to three parts. To interpret the hydrological signal, we used a new metric, the relative magnitude of net increased river extents, to analyze the dominant signal and retrieve hotspots of river changes. The original three stories of river changes (widening in the third-pole and pan-arctic, narrowing in arid regions of interior continents, and the Yellow River case) are reorganized into the third-part – hotspot analysis. In addition, we conducted a systemic climate change analysis for the eight hotspots to discuss the potential forcings. Due to updates with SWORD and the hierarchical analysis, the specific statistics of river changes are somehow different. Nevertheless, the main results and arguments remain similar in general. The renewed results explicitly show where the widening/narrowing signal has been prominent and what the

potential role of climate change and human interferences is. **Given revision in methods and analysis, as well as comments from the reviewers, we modified the title to better reflect the main idea of this work.**

Details of these changes are specified as follows.

(1) New information about morphological dynamics

This new information identifies river basins (Level-9 in the HydroBASINS dataset) where significant river extent changes reflect morphological dynamics (migration or shifting of meandering/multi-thread channels). Such signals can be distinguished based on the OCI map and the morphological attributes. We developed an expert-based guidance to manually interpreted this type of signal (Fig. S3, S4) for training samples and optimized three widely-used machine learning approaches for the global classification.

(2) Separated evaluation of the dam-related river expansion

We retrieved new reservoir-type river reaches by using an updated global reservoir inventory. The results separately evaluate the dam-related river expansion in the second part. We focus on where new dams were built and to what extent the dams impacted river water extent changes.

(3) Hotspot analysis of the hydrological signal on river extent changes

In addition to a general description of the pattern of river extent changes, we performed further hotspot analysis to identify the location and extent of the change signal. Given the high spatial variability of such a signal, we focused on the four largest river-widening hotspots and the four largest narrowing hotspots. Different climatic datasets were used to analyze the general trends of precipitation and evaporation changes in these hotspots. The climate analysis generally explains the contrasting pattern of river extent changes reflected in the two types of hotspots, except the Yellow River basin, where improvement of water use policy probably played a critical role. We argue that the increasing human water extraction could have aggravated the drying-related river narrowing in the four negative hotspots.

(4) Uncertainty analysis

To address the reviewers' concerns about the OCI-based statistics' reliability, we addressed the algorithm robustness, the seasonal effect, and the uneven coverage of the Landsat observations in a new section in the Methods. We compared different sources of data on a couple of typical cases to evaluate the robustness of the Landsat-based river change patterns, and demonstrated that seasonal effects had been primarily mitigated in the statistics algorithm.

Finally, we compared in-situ discharge observations with our statistics to show that the metric used in this study is a generally reliable indicator of the hydrological signal. We also discussed that uncertainties are mainly sourced from the temporal unevenness of Landsat observations which means temporal ambiguity for a few local regions.

(5) Figures/Tables/supplementary files

Figures were reorganized and synthesized. Some zoom-in maps and figures in the supplementary files were merged with the figures in the main text. We supplemented new figures supporting the updated analysis and methods in the main text and supplementary files.

(6) Language and others

We would like to thank the reviewers for the careful reviewing and detailed comments. We corrected presentation problems and grammatical errors, e.g., region names and use of some terms and phrases. In addition, new references were added to support some of the new arguments. We also included the latest literature relevant to this work for comparison.

We attach the detailed item-by-item response to all comments and suggestions for the evaluation.

Sincerely

Chungiao Song

Corresponding author and other co-authors

REVIEWERS' COMMENTS

Reviewer #1 (Remarks to the Author):

As reviewer #1, I reviewed this MS for this second run. I am pleased to say that the authors carefully took into account the different suggestions. We have now a very impressive MS with clear messages. The different drivers are convincingly introduced and separated and the take-home messages are straightforward. I am very pleased to recommend it for publication. I have only very minor comments/suggestions in the joined annotated files.

Reviewer #3 (Remarks to the Author):

The authors are commended for the considerable revision work. The paper has become more focused, and the process distinctions are very helpful. While I'm still not entirely clear on precisely what is new in the paper, the three process categories (R,M,H) are new, and provide a helpful lens for interpreting the width changes. The statistics on river widening and narrowing have been reported in other work, though perhaps not with the detail and process distinction that is offered here. I do question whether one can neatly partition these three without overlap, but I would agree there is probably a suitably dominant category for any given river. The important point is made (l.147-148) that there can be multiple processes at work.

I think the identification of the global hotspots of change is the most notable part of the work. In my review of the earlier version, I pointed to the problem of fairly broad and diffuse statements about river change - I think the hotspots component could be a central focusing research question, and it should perhaps be duly reflected in the title. I would say the abstract does not effectively capture this contribution (the conclusion does this better, to some extent), and it could be sharpened quite a bit to reflect this focus, and capture reader attention. While the broadscale quantification of river change is impressive, there is important context and purpose here that makes this more than an accounting exercise.

The M-Type section does not come away with any new, tangible conclusions (as stated [l.153], it would be difficult to do so!). The importance here is mainly partitioning this population from the larger dataset - if more room was needed for the hotspot discussion, this could be tightened a bit. I find the link between stable morphologies and socio-economic development (l.344ff) a bit.. fraught, as built-up areas are generally sited far from more dynamic (i.e. steepland, mountain front) rivers and fans with higher sediment supply and more active meander dynamics. The countries in the ranking with the most stable rivers (Fig 8) are also the ones in old, flat and stable cratonic settings (excepting Japan, of course!). But indeed, embankments do keep things static, as well.

The figures are very nicely done, and convey all the key messages effectively. I have not thoroughly checked the text, but I have listed a few issues, below. With some further streamlining, I expect this paper will be of great interest to Nature Geoscience readership.

l.28 "water occurrence changes" - suggest "changes in water extent"

l.37 "could be likely driven" - suggest "could likely be driven"

l.38, 45, 56 simply, "interference" (no s)

l.49 "Rivers are one of.."

l.56 - when a river becomes a reservoir, is it 'widened', or has it become a lake (i.e. not a river anymore)?

l.74-75 "..changes exclusively in global rivers.." Why exclusively?

l.126 "On this type of river basins.." remove s, or make it "these types of river basins"

l.128 Are these necessarily flow regime changes, or could they be evolving morphologies?

l.138 Similarly, is there necessarily decreased channel stability, or is that how the river evolves?

l.154 in-situ, or on-site.

l.235 "outpowering that of a decrease" - reword.

l.303 "hotspot (d)"

I.321 "..(less than.." No capital here.
I.327 "dry-up situation"?

REVIEWER COMMENTS

Reviewer #1 (Remarks to the Author):

As reviewer #1, I reviewed this MS for this second run. I am pleased to say that the authors carefully took into account the different suggestions. We have now a very impressive MS with clear messages. The different drivers are convincingly introduced and separated and the take-home messages are straightforward. I am very pleased to recommend it for publication. I have only very minor comments/suggestions in the joined annotated files.

Response: Thanks for the encouraging comments. We sincerely appreciate all your constructive comments that significantly improved the rigor and quality of this work.

Line2: "...global rivers" ---"...in rivers at global scale"

Response: We now followed your and the reviewer #3's comments, and revised the title to "Satellites reveal hotspots of global river extent change".

Line145: "...low hydraulic efficiency, high flow velocities..." ---it is contradictory.

Response: Sorry for the mistakes here. We checked the references in the context and removed the term 'high flow velocities'. This sentence was rephrased for clarity.

Line145-146: "...intense erosion and heavy sediment deposition..." ---I would just say: low hydraulic efficiency partly due to tectonic activities and high sediment delivery

Response: We have reworded the sentence as suggested, which is also quoted below "...which is probably attributed to low hydraulic efficiency partly due to high sediment delivery, tectonic activities and intense erosion".

Some comments in the Supplementary Text:

Line27: "...braiding, and meandering)..." ---this is not really a change?!

Response: To be accurate, we revised this sentence to 'we selected fifteen cases covering different types of river extent changes (e.g., stable, widening, narrowing) and morphological dynamics of braiding or meandering rivers within different climate zones'.

Line91: "...High..." ---low?

Response: We checked that the use of 'high' is correct here. Aridity Index is defined as the ratio of precipitation over potential evaporation, and thus high aridity index often indicates humid climate according to this definition.

Line93-94: "Light blue circles represent the positive hotspots, and light-yellow circles represent the negative hotspots" ---where? => areas you mean rather than circles?

Response: We thank the reviewer for pointing out this, and are sorry for our overlook previously. The sentence referred here aims to explain figure plots in panel (b) rather than

panel (a). We thus accordingly moved this sentence to the later part, and revised the sentence to ‘Light blue plots represent positive hotspots, and light-yellow plots represent negative hotspots.’

Line94: “...aridity...” ---?

Response: We modified the figure title for clarity, which is also quoted here; ‘Statistics of basin-wide mean aridity in the eight hotspots of significant river changes.’

Line95: “...are characterized by relatively lower aridity index...” ---I don't really see that on b) figure.

Response: The differences in aridity levels between the negative and positive hotspots are not visually prominent because of the exceptions of the region (g) and (b). The main text gives details about the comparisons, and we deleted this sentence here.

Reviewer #3 (Remarks to the Author):

The authors are commended for the considerable revision work. The paper has become more focused, and the process distinctions are very helpful. While I'm still not entirely clear on precisely what is new in the paper, the three process categories (R,M,H) are new, and provide a helpful lens for interpreting the width changes. The statistics on river widening and narrowing have been reported in other work, though perhaps not with the detail and process distinction that is offered here. I do question whether one can neatly partition these three without overlap, but I would agree there is probably a suitably dominant category for any given river. The important point is made (l.147-148) that there can be multiple processes at work.

Response: Thank you for your overall positive evaluations. We agree with the reviewer regarding challenges in partitioning different types of river change signals, and our interpretation of the results has taken this into account as the reviewer echoed. Relevant statements were polished in the abstract and discussion sections to highlight the novelty of this work.

I think the identification of the global hotspots of change is the most notable part of the work. In my review of the earlier version, I pointed to the problem of fairly broad and diffuse statements about river change - I think the hotspots component could be a central focusing research question, and it should perhaps be duly reflected in the title. I would say the abstract does not effectively capture this contribution (the conclusion does this better, to some extent), and it could be sharpened quite a bit to reflect this focus, and capture reader attention. While the broadscale quantification of river change is impressive, there is important context and purpose here that makes this more than an accounting exercise.

Response: We appreciate the reviewer's excellent advice on the title and the abstract. The title was now accordingly modified to highlight the hotspots analysis. Taking this suggestion and word limit into account, we modified the abstract section to reflect the novel finding of this work, especially those related with the hotspot analysis. We moved some details of results originally in the abstract section to now in the conclusion section.

The M-Type section does not come away with any new, tangible conclusions (as stated [l.153], it would be difficult to do so!). The importance here is mainly partitioning this population from the larger dataset - if more room was needed for the hotspot discussion, this could be tightened a bit. I find the link between stable morphologies and socio-economic development (l.344ff) a bit.. fraught, as built-up areas are generally sited far from more dynamic (i.e. steepland, mountain front) rivers and fans with higher sediment supply and more active meander dynamics. The countries in the ranking with the most stable rivers (Fig 8) are also the ones in old, flat and stable cratonic settings (excepting Japan, of course!). But indeed, embankments do keep things static, as well.

Response: Thanks for pointing out the concerning items regarding the M-Type and stable type sections. We moderately modified the M-Type section for a precise and clear focus. Regarding the reason for river extent stability in built-up areas, our revised discussion includes the possibility that built-up areas tend to avoid high-dynamic regions. This perspective was also reflected in the abstract and conclusion sections.

The figures are very nicely done, and convey all the key messages effectively. I have not thoroughly checked the text, but I have listed a few issues, below. With some further streamlining, I expect this paper will be of great interest to Nature Geoscience readership.

Response: Thanks for the encouraging comments and careful reading. We carefully considered the listed points and made corrections accordingly.

Line28: "...water occurrence changes..." --- suggest "changes in water extent"

Response: Revised the sentence as suggested.

Line37: "...could be likely driven..." --- suggest "could likely be driven"

Response: Revised the sentence as suggested.

Line38, 45, 56: simply, "interference" (no s)

Response: Revised the sentence as suggested.

Line49: "Rivers as one of..." ---"Rivers are one of..."

Response: Revised the sentence as suggested.

Line56: "...substantially widening the natural wetted river channels." --- when a river becomes a reservoir, is it 'widened', or has it become a lake (i.e. not a river anymore)?

Response: Generally, reservoirs are built upon certain sections of the river reaches and become part of the river courses, given that river water still runs through them, although in an unnatural way (artificially controlled). In this perspective, reservoirs, unlike lakes, cannot be seen as isolated from the river network in general.

Line74-75: "...changes exclusively in global rivers..." ---Why exclusively?

Response: We originally meant that river extent changes at a global scale have not been adequately investigated. The term 'exclusively' was removed and the statement was revised for clarity.

Line126: "On this type of river basins..." --- remove s, or make it "these types of river basins"

Response: Thanks for the correction. Revised the sentence as suggested.

Line128: "...flow regime changes of meandering, braided, anabranching or wandering river channels" ---Are these necessarily flow regime changes, or could they be evolving morphologies?

Response: We agree that both cases could be true. To be accurate, we reworded the term 'changes' as 'variations' in this sentence.

Line138: "...reflect decreased channel stability or a highly active state of river channel evolution" ---Similarly, is there necessarily decreased channel stability, or is that how the river evolves?

Response: By stating 'a highly active state of river channel evolution' we mean that the river behavior could reflect a particular state of rivers that could be natural. We later discussed that the forcings of such behavior could be complicated and challenging to identify at this stage. This statement was rephrased for clarity.

Line154: "...in-site..." ---in-situ, or on-site.

Response: Thanks for the point. Revised the sentence as suggested.

Line235-236: "...outpowering that of a decrease" --- reword.

Response: This sentence was rephrased as ‘the majority of the mega basins (20 out of the 25) are positive, with increases greater than decreases.’

Line303: "...hotspot d..." ---"hotspot (d)"

Response: Thanks for the point. Revised the sentence as suggested.

Line321: "(Less than 1 m³/s runoff" ---"... (less than..." No capital here.

Response: The mistake was corrected accordingly.

Line327: "dry-up situation"?

Response: The term (dry-up) was corrected to ‘dried-up’. We also performed a careful check throughout the full text and made all the relevant correction.